# Intimate partner violence is a barrier to antiretroviral therapy adherence among HIV-positive women: Evidence from government facilities in Kenya

Bornice C. Biomndo[1]⊚*, Alexander Bergmann[2]⊚, Nils Lahmann[3], Lukoye Atwoli[4,5]⊚

1 Institute of Health and Nursing Science, Charité—Universitätsmedizin Berlin, Berlin, Germany, 2 Department of Biology, Universität Leipzig, Leipzig, Germany, 3 Clinic for Geriatrics and Geriatric Medicine, Charite-Universitätsmedizin Berlin, Berlin, Germany, 4 Department of Mental Health, Moi University School of Medicine, Eldoret, Kenya, 5 Medical College East Africa, Aga Khan University, Nairobi, Kenya

⊚ These authors contributed equally to this work.
* bornice.biomndo@charite.de, bbc_j@yahoo.co.uk

**Data Availability Statement:** We have uploaded our data set to Zenodo and it is available via the following URL: https://zenodo.org/record/

## Abstract

### Introduction

Intimate Partner Violence (IPV) is linked to low engagement with HIV management services and adverse clinical outcomes, including poor ART adherence. In sub-Saharan Africa, studies on pregnant/postpartum women and transactional sex workers have produced divergent evidence regarding IPV's association with poor ART adherence. We investigate this association among a broad group of women.

### Methods

We sampled 408 HIV-positive women receiving free ART from different types of HIV clinics at government health facilities, assessing for IPV exposure by a current partner, ART adherence rate, and other factors that affect ART adherence (e.g. education, disclosure). ART adherence rates were measured using the Visual Analogue Scale (VAS); responses were dichotomised at a $\geq$95% cut-off. Multiple logistic regression models assessed the association between the independent variables and ART adherence.

### Results

The participants' mean age was 38.6 (range: 18–69 years). The majority had ever attended school (94%, $n = 382$), were in monogamous marriages (70%, $n = 282$), and had disclosed status to partners (94%, $n = 380$). Overall, 60% ($n = 242$) reported optimal ART adherence ($\geq$ 95%) in the previous 30 days. The prevalence of IPV by the current partner was 76% ($CI95 = 72$–80%). Experiencing physical IPV (AOR 0.57, $CI95$: 0.34–0.94, $p = .028$), sexual IPV (AOR 0.50, $CI95$: 0.31–0.82, $p = .005$), or controlling behaviour (AOR 0.56, $CI95$: 0.34–0.94, $p = .027$) reduced the odds of achieving optimal adherence, while a higher education level and having an HIV-positive partner increased the odds.

4135394#.YGNt469Kg2x (DOI: 10.5281/zenodo. 4135394).

**Funding:** The authors received no specific funding for this work.

**Competing interests:** The authors have declared that no competing interests exist.

## Conclusion

IPV is common and is associated with suboptimal ART adherence rates among a broad group of HIV-positive women. ART programs could consider incorporating basic IPV interventions into regular clinic services to identify, monitor and support exposed women, as they might be at risk of poor ART adherence. Still, there is need for more research on how IPV affects ART adherence.

## Introduction

Intimate Partner Violence (IPV) is recognised as a factor behind low uptake and engagement with HIV management services [1–7]. IPV is defined as acts of physical aggression, sexual coercion, psychological/emotional abuse, or controlling behaviour by a former or current intimate partner [8]. Studies in resource-rich settings such as the United States of America and Canada estimate that 68–95% of HIV-positive women experience IPV [1,3]. Estimates available from sub-Saharan Africa report that the prevalence of the various forms of IPV among HIV-positive women ranges from 26% to 72% [3,9–11]. Studies from East Africa indicate that HIV-positive women are 2–10 times more likely to report lifetime IPV than their HIV-negative counterparts [12–14]; one in three HIV-positive women experiences IPV [10]. In Kenya, population-based surveys indicate that, among women aged 15–49 years, the national IPV prevalence rate is 47% [15], while the national adult (≥15 years old) HIV prevalence rate is 5.2%, which is slightly higher than that of men (4.5%) [16].

Several studies that include women from sub-Saharan Africa have established a significant bidirectional association between IPV and positive HIV serology [1,4,12–14,17–19]. IPV is a risk factor for HIV infection among women [18,20] since women who experience it are more likely to be exposed to risky sexual behaviour, violent sexual acts, and forced sex [21,22]. Moreover, women who experience IPV are rarely in a position to negotiate for condom use and generally have reduced access to HIV testing or healthcare services [1,6,12,23]. Experiencing IPV can also result from a woman revealing her HIV-positive status (post-disclosure IPV). Reports indicate that a significant number of HIV-positive women experience IPV as a result of behaviour changes resulting from an HIV-positive diagnosis, a disclosure of positive status, or attempts to discuss HIV testing and treatment options [1,7,10,13,23–26]. Actual IPV and fear of IPV is associated with non-disclosure, poor mental health, missing clinic appointments, and prioritising safety over medication compliance, which adversely affect HIV prevention and ART initiation, adherence, and retention strategies [1,6,19,25,27–29].

Studies that explore the role of IPV in uptake and compliance with HIV management services link IPV to poor service uptake, poor medication adherence, and consequent virological failure, higher mortality, and increased episodic diseases [1,28,30]. In 2015, Hatcher et al. [1] noted in their systematic review that these findings were skewed towards resource-rich settings. A knowledge gap existed regarding this association in sub-Saharan Africa, which contains an estimated 12 million out of 17 million people receiving ART globally [31] and where IPV and HIV infection are prevalent and disproportionately affect women [1,6,32,33]. Recent studies from sub-Saharan Africa have aimed to fill this knowledge gap, yielding contrasting and inconclusive evidence. While some studies have established exposure to IPV as a risk factor for poor ART adherence among women [19,23] or associated IPV with reduced odds of ART adherence [7], others have found no significant association [34,35]. So far, the evidence from sub-Saharan Africa has been based on research among key populations, such as

transactional sex workers and women at Prevention of Mother-to-Child Transmission (PMTCT) clinics. However, to determine whether IPV exerts an overarching influence on ART adherence, the association between both variables must be explored among a broader sample of HIV-positive women with diverse socioeconomic characteristics who are receiving the available standard ART from different types of HIV clinics.

In this observational study, we sought to determine the prevalence of IPV among HIV-positive women and explored the relationship between IPV exposure and ART adherence rates. We investigate whether physical IPV, sexual IPV, emotional IPV, and controlling behaviour are associated with self-reported suboptimal adherence among a sample of HIV-positive women (n = 408) receiving free ART at government facilities in western Kenya.

## Conceptual framework

This study was guided by a combination of two conceptual frameworks. Firstly, we used the theoretical framework on ART adherence which stipulates that there are sociodemographic drivers which alongside treatment regime, provider-patient relationship, clinic setting, and disease characteristics, affect a patient's level of medical adherence [36–38]. More specifically, that there are socio-cultural and interpersonal factors (i.e. partner interference) which physically or psychologically undermine a woman's ability or motivation to adhere to ART [1,29,38–41]. Secondly, the WHO conceptual framework on the adverse health effects of IPV on women provides a structure for understanding the possible role IPV has in poor ART adherence. There are three key mechanisms and pathways: physical trauma, psychological trauma/stress, and fear and control. These can directly or indirectly lead to injury, mental health problems, limited control of one's health and limited health care seeking behaviour [42], which could consequently interfere with ART adherence.

## Materials and methods

### Study design and setting

This cross-sectional survey was conducted in March and April 2018 at twelve HIV clinics in government health facilities in Kenya. The HIV clinics are part of the AMPATH (Academic Model Providing Access to Healthcare) partnership that cooperates with the Ministry of Health and a consortium of universities to offer free HIV care and treatment services [43]. At the time of the survey, there were 79,728 HIV-positive women (aged ≥15 years) enrolled in the ART program; 40,370 were actively receiving ART [44]. We selected six urban and six rural clinics for the survey based on their number of active HIV-positive women receiving ART; the clinics were chosen to account for the socioeconomic diversity of the counties served by the AMPATH partnership. We included large clinics with many women active on ART, as well as smaller health facilities that we purposively selected to represent less populous communities.

### Study population

The study population consisted entirely of HIV-positive women who were actively receiving free ART at the HIV clinics. Eligible respondents were at least 18 years old, currently in an intimate partner relationship, and had begun ART at least six months before the survey. We set a minimum duration of six months since beginning ART in an effort to reduce factors affecting adherence that are related to the ART initiation. Additionally, according to the Kenya national ART guidelines, it is expected that at six months the clients should be compliant since they have sufficient understanding of HIV, medication dosage, their clinical progress, and the importance and benefits of ART adherence [45].

## Sample size determination

The sample included 408 women. The sample size was calculated through a power proportion test [46] in R (stats package, R Core Team 2013), using the national IPV prevalence rate of 47% [47] as the proportion, at a power of 0.8 and a significance level of 0.05, and with the aim of conducting a logistic regression. Due to feasibility considerations, we decided to use the prevalence rate of physical violence (45%) as the proportion, in order to get a workable sample size and because it was the most reported form of violence among women [47]. Proportional stratified sampling was applied to determine the number of women to sample from each of the selected clinics. That is, the number of women active on ART at each clinic was weighed against the total number of women on ART at AMPATH to determine each clinic's share of the total sample.

## Data collection

**IPV measurement.**   Exposure to physical IPV, sexual IPV, emotional IPV, and controlling behaviour by a current intimate partner was measured using the Demographic Health Survey (DHS) module on Domestic Violence. This module is a modified version of the Conflict Tactics Scale by Murray A. Straus and is widely used to measure spousal violence within the household context [47–49]. An intimate partner is defined as someone to whom the woman was currently married (whether in a monogamous or polygamous marriage) or with whom she was in a romantic relationship. The DHS questions on IPV depict specific forms of physical IPV (e.g. slapping, kicking), sexual IPV (i.e. use of physical force or threats to have sexual intercourse), emotional IPV (e.g. humiliation, insults), and controlling behaviour (hindrance of social contact). Each can be answered with 'No' or 'Yes'. A 'Yes' response to any question was considered to constitute exposure to IPV.

The DHS module on domestic violence measures both lifetime IPV and IPV within the previous 12 months by a current or former partner. By recruiting only women who were currently in a relationship, and slightly altering the DHS questions from '*Did your (last) husband/partner ever. . .*' to '*Did your husband/partner ever. . .*', we focused on exposure to IPV by a current partner within the lifetime of the ongoing relationship (relationship-specific IPV). Our interest was investigating whether being in an environment where IPV occurs affects the ability to adhere to ART.

**ART adherence measurement.**   To measure ART adherence, we used the AIDS Clinical Trials Group (ACTG) Adherence Follow-up Questionnaire and the Brief Adherence Self-report. The latter contains a 30-day Visual Analogue Scale (VAS) commonly used in resource-limited settings because it is practical, easy to administer, inexpensive, and does not require high literacy levels [7,50–53]. The participants were asked to best estimate the percentage of ARV dosage they took in the last 30 days by marking an X or O on the VAS line measuring from 0% to 100%. Selecting 0 indicated that they had taken none of the prescribed drugs; 100% meant they had not missed a single dose.

The DHS module on Domestic Violence and the ACTG Adherence Follow-up Questionnaire and Brief Adherence Self-report are validated measurement tools that have been used in similar populations and settings [7,31,32,54–56]. Moreover, the ACTG Follow up and Self-report measures are one of several ways that the HIV clinics monitor client ART adherence rates; therefore, they were already familiar to both the recruiters and the participants.

For the other covariates, we used a broad set of socio-economic drivers which existing literature identifies as potentially affecting ART adherence: age, marital status, length of time on ART, education, income, HIV status disclosure, partner's HIV status, social support from the partner (if the partner is involved in the woman's ART), the partner's alcohol consumption,

and the area of living. The IPV and ART subscales and the questions concerning potentially relevant variables were combined into a four-page questionnaire administered in paper form.

## Procedure

Participants were recruited from mixed HIV clinics (non-specialised clinics where female and male HIV-positive adults are reviewed and given medication refills), PMTCT clinics, maternal and child health clinics, and express clinics (for clients who are categorised as stable/virally suppressed, who therefore receive drug refills without rigorous review by a healthcare provider). The clinical officers and nurses who provide and supervise ART during regular clinical care visits recruited the participants. They were trained on the aim of the study, the tools, and the recruitment process (random selection, eligibility criteria). Randomised lists were created using Microsoft Excel, considering the number of clinical officers/nurses administering the questionnaires per clinic and the estimated number of female clients the healthcare provider attends to per day (retrieved from the daily clinic patient lists). Each healthcare provider received a randomised list which they used to recruit at least five women per day from their daily patient lists.

On completion of the routine clinical check-up in the regular clinic examination rooms, women whose session coincided with the random number from the list and who fit the study inclusion criteria were informed about the survey and asked if they were willing to participate. Prior to this, if the woman was accompanied by another person or a child who was old enough to understand the conversation and old enough be left alone, the person/child was politely asked to leave the examination room. The women were assured that participation or non-participation would not affect access to treatment. After they consented to participation, an informed consent form was provided, which they signed before the questionnaire was administered. Assistance was provided for those who could not read, needed clarification, or preferred that the questionnaire be administered as an interview. All of the consent forms and questionnaires were available in English and Kiswahili. Unfortunately, some of the rooms at the clinics were shared; healthcare providers were responsible for ensuring maximum possible privacy by, for example, asking colleagues to temporarily leave the room or by moving the desk or drawing a curtain. On completion of the survey, the women were given financial compensation (KSh 100) for their participation; this was referred to as 'transport money'. Women who reported exposure to IPV were offered a list of places where they could receive free social and legal assistance within the health facility. The list also contained local government authorities, or non-governmental organisations in the area which were involved in gender-based violence prevention/women empowerment work.

## Data analysis

Data from the questionnaires were entered and imported into R (Version 1.2.1335) for analysis. The responses regarding exposure to IPV were combined into five new variables: physical IPV, sexual IPV, emotional IPV, controlling behaviour, and overall exposure to IPV. If a woman answered 'Yes' to any of the questions on exposure to physical IPV, she was coded as '1'; if not, she was coded as '0'. The same was done for the other forms of IPV; overall exposure to IPV meant that the woman answered 'Yes' to any form of IPV.

Since our hypothesis was on relationship-specific IPV, our main analysis was based on exposure of IPV at any time within the relationship. However, to identify the possible impact of this decision on our results, we repeated the modelling procedure, with only women who were exposed to IPV in the last 12 months.

The responses to the VAS were used as the ART adherence outcome. Participants' answers were dichotomised using the conservative optimal ART adherence level of ≥95% and suboptimal adherence of <95%, which was suggested by Paterson et al. [57] as necessary for achieving HIV viral suppression. We also performed a second analysis with a lower cut-off of ≥85% for comparison. Because both analyses yielded comparable results, we present analyses based on the more conservative cut off. We checked data quality and uni- and bivariate distributions using descriptive data analysis techniques. Since the questionnaire contained intimate questions that the participants could potentially skip, a missing value analysis was performed. None of the variables had more than 5% missing values. Additionally, no systematic relationships between the missing values were detected.

We used simple logistic regression models to examine the relationships between each of the independent variables and the dependent variable (ART adherence). Next, ART adherence was regressed stepwise for each of the independent variables. A possible limitation of regressing ART adherence on all independent variables together is that it prevents the identification of suppressor/moderator effects and runs the risk of overfitting the model. Assumptions were checked before conducting all logistic regression analyses; log likelihood-based Pseudo $R^2$ measures and AIC scores were used to evaluate goodness of fit. Due to multicollinearity between the IPV variables, we decided to report four multiple logistic regression models, each of which includes only one IPV variable alongside the other independent variables. Despite the nested structure of our data (multiple women clustered in each of the 12 clinics), we decided against a multilevel modelling procedure. The low intraclass correlation coefficient of ICC = .15 that we derived from an unconditional multilevel logistic regression implies that only 15% of the individual variation in the underlying propensity of low ART uptake is due to systematic differences between the clinics [58,59]. Additionally, with as few as 12 clusters, fixed-effect estimates associated with level-2 predictors could have been severely biased [60,61]. We also did not include clinics as a fixed effect in our main analysis because adding 12 additional dummy coded variables to the analysis would have led to predictor combinations with very few to zero observations. However, we performed a sensitivity analysis in which clinic was added as a predictor variable. We identified significant differences among the clinics but there were no significant changes in the respective model parameter estimates from previous models (S3 Table).

## Ethical considerations

Ethical approval was granted by the Moi University/Moi Teaching and Referral Hospital Institutional Research and Ethics Committee (IREC). Consent to participate was established through a written Informed Consent Form that the women signed.

## Results

### ART adherence and the prevalence of IPV in the current relationship

As shown in Table 1, the mean age of the participants was 38.6 years (range: 18–69 years old) and average time on ART was 78.8 months. The majority (94%, $n = 382$) had ever attended school, although only 13% had an education higher than secondary school ($n = 49$). Most women were in monogamous marriages (70%, $n = 282$), had disclosed their HIV-positive status to their partners (94%, $n = 380$), and knew that their partner was HIV-positive (64%, $n = 260$) (Table 1). Overall, 60% ($n = 242$) reported achieving optimal adherence (≥ 95%) in the last month. A large majority (76%, $CI95 = 72–80%$) of the women reported exposure to a form of IPV from their current partner. Of those exposed to IPV, 75% ($CI95 = 71–79%$) had experienced emotional IPV; 70% had experienced physical IPV ($CI95 = 66–74%$); 49%

**Table 1. Participant's demographics, total and relative frequencies of IPV, and other variables stratified by optimal or suboptimal ART adherence.**

| Independent Variable | ART Adherence | | | | | |
| --- | --- | --- | --- | --- | --- | --- |
| | Total | | Suboptimal (<95%) | | Optimal (>95%) | |
| | Mean | SD | Mean | SD | Mean | SD |
| Age in years | 38.6 | 8.5 | 37.7 | 8.5 | 39.2 | 8.4 |
| Length of time on ARVs in months | 78.8 | 48.5 | 73.3 | 44.4 | 81.9 | 50.5 |
| | n | % | n | % | n | % |
| Physical IPV No | 189 | (46.7) | 62 | (38.0) | 127 | (52.5) |
| Yes | 216 | (53.3) | 101 | (62.0) | 115 | (47.5) |
| Sexual IPV No | 255 | (63.0) | 86 | (52.8) | 169 | (69.8) |
| | 150 | (37.0) | 77 | (47.2) | 73 | (30.2) |
| Emotional IPV No | 170 | (42.0) | 55 | (33.7) | 115 | (47.5) |
| Yes | 235 | (58.0) | 108 | (66.3) | 127 | (52.5) |
| Controlling Behaviour No | 286 | (70.6) | 100 | (61.3) | 186 | (76.9) |
| Yes | 119 | (29.4) | 63 | (38.7) | 56 | (23.1) |
| Education None | 24 | (5.9) | 13 | (7.9) | 11 | (4.5) |
| Primary | 224 | (55.2) | 87 | (53.0) | 137 | (56.6) |
| Secondary | 109 | (26.8) | 55 | (33.5) | 54 | (22.3) |
| Tertiary | 49 | (12.1) | 9 | (5.5) | 40 | (16.5) |
| Marital Status In a relationship | 33 | (8.1) | 17 | (10.4) | 16 | (6.6) |
| Monogamous marriage | 282 | (69.6) | 102 | (62.6) | 180 | (74.4) |
| Polygamous marriage | 90 | (22.2) | 44 | (27.0) | 46 | (19.0) |
| Disclosure No | 25 | (6.2) | 14 | (8.6) | 11 | (4.5) |
| Yes | 380 | (93.8) | 149 | (91.4) | 231 | (95.5) |
| Partner (HIV-positive) No | 145 | 35.8 | 69 | (42.3) | 76 | (31.4) |
| Yes | 260 | (64.2) | 94 | (57.7) | 166 | (68.6) |
| Partner accompanies to clinic/reminds to take ARV No | 176 | (43.7) | 84 | (52.2) | 92 | (38.0) |
| Yes | 227 | (56.3) | 77 | (47.8) | 150 | (62.0) |
| Partner is drunk Never | 210 | (51.8) | 73 | (44.8) | 137 | (56.6) |
| Sometimes | 121 | (29.9) | 58 | (35.6) | 63 | (26.0) |
| Often | 74 | (18.3) | 32 | (19.6) | 42 | (17.4) |
| Woman in violence No | 345 | (85.2) | 128 | (78.5) | 217 | (89.7) |
| Yes | 17 | (4.2) | 9 | (5.5) | 8 | (3.3) |
| Fights back | 43 | (10.6) | 26 | (16.0) | 17 | (7.0) |
| Area Rural | 163 | (40.2) | 80 | (49.1) | 83 | (34.3) |
| Urban | 242 | (59.8) | 83 | (50.9) | 159 | (65.7) |

Note. SD = Standard Deviation.

reported sexual IPV (*CI95* = 44–54%); and 39% (*CI95* = 34–44%) reported controlling behaviour from their current partner. When narrowed down to IPV exposure in the last 12 months, 72% (*CI95* = 68–76%) had experienced emotional IPV; 53% had experienced physical IPV (*CI95* = 48–58%); 40% reported sexual IPV (*CI95* = 35–45%); and 45% (*CI95* = 40–50%) reported controlling behaviour.

## Logistic regression of ART on IPV

Simple regression analyses of the IPV variables and ART adherence revealed that all forms of IPV were significantly correlated to suboptimal ART adherence. As shown in Table 2, when adjusted for other factors affecting ART adherence, physical IPV, sexual IPV and controlling

**Table 2. Multiple regression models of factors associated with ART adherence.**

| Predictor Variable | | Model 1 | | | Model 2 | | | Model 3 | | | Model 4 | | |
| --- | --- | --- | --- | --- | --- | --- | --- | --- | --- | --- | --- | --- | --- |
| | | AOR | 95% CI | p | AOR | 95% CI | p | AOR | 95% CI | p | AOR | 95% CI | p |
| **Physical IPV** | (None) | | | | | | | | | | | | |
| | Yes | 0.57 | 0.34–0.94 | .028* | | | | | | | | | |
| **Sexual IPV** | (None) | | | | | | | | | | | | |
| | Yes | | | | 0.50 | 0.31–0.81 | .005** | | | | | | |
| **Emotional IPV** | (None) | | | | | | | | | | | | |
| | Yes | | | | | | | 0.65 | 0.39–1.10 | .108 | | | |
| **Controlling Behaviour** | (None) | | | | | | | | | | | | |
| | Yes | | | | | | | | | | 0.56 | 0.34–0.94 | .027* |
| **Age** | | 1.02 | 0.99–1.05 | .145 | 1.02 | 0.99–1.05 | .129 | 1.02 | 0.99–1.05 | .140 | 1.02 | 0.99–1.05 | .191 |
| **TARV** | | 1.00 | 0.99–1.00 | .529 | 1.00 | 0.99–1.00 | .556 | 1.00 | 0.99–1.00 | .499 | 1.00 | 0.99–1.00 | .662 |
| **Education** | (None) | | | | | | | | | | | | |
| | Primary | 2.23 | 0.80–6.26 | .124 | 2.16 | 0.78–6.03 | .136 | 2.09 | 0.76–5.77 | .151 | 2.01 | 0.73–5.52 | .173 |
| | Secondary | 1.29 | 0.44–3.81 | .640 | 1.30 | 0.44–3.80 | .635 | 1.22 | 0.42–3.55 | .713 | 1.24 | 0.43–3.61 | .687 |
| | Tertiary | 5.17 | 1.12–25.67 | .039* | 5.71 | 1.25–28.10 | .028* | 4.48 | 0.99–21.81 | .560 | 4.64 | 1.02–22.58 | .050* |
| **Marital Status** | (In a relationship) | | | | | | | | | | | | |
| | Monogamous marriage | 1.79 | 0.77–4.17 | .175 | 1.44 | 0.62–3.79 | .387 | 1.65 | 0.71–4.00 | .237 | 1.63 | 0.70–3.76 | .248 |
| | Polygamous marriage | 1.03 | 0.41–2.60 | .989 | 0.89 | 0.35–2.21 | .797 | 0.94 | 0.38–2.34 | .900 | 0.95 | 0.38–2.35 | .904 |
| **Area** | (Rural) | | | | | | | | | | | | |
| | Urban | 1.31 | 0.81–2.12 | .271 | 1.31 | 0.81–2.13 | .268 | 1.28 | 0.79–2.07 | .317 | 1.276 | 0.79–2.07 | .320 |
| **Partner's Alcohol Consumption** | (None) | | | | | | | | | | | | |
| | Sometimes | 0.71 | 0.42–1.20 | .202 | 0.68 | 0.40–1.15 | .150 | 0.72 | 0.43–1.23 | .233 | 0.67 | 0.40–1.140 | .140 |
| | Often | 1.60 | 0.79–3.33 | .199 | 1.85 | 0.76–1.37 | .240 | 1.57 | 0.78–3.25 | .215 | 1.44 | 0.72–2.94 | .315 |
| **Partner's HIV Status** | (Negative) | | | | | | | | | | | | |
| | Positive | 1.70 | 1.03–2.81 | .037* | 1.69 | 1.03–2.80 | .039* | 1.60 | 0.97–2.62 | .064 | 1.60 | 0.97–2.63 | .062 |
| **Supporting partner** | (No) | | | | | | | | | | | | |
| | Yes | 1.24 | 0.75–2.06 | .398 | 1.23 | 0.74–2.03 | .434 | 1.23 | 0.74–2.03 | .431 | 1.28 | 0.77–2.13 | .331 |
| **Woman is Violent** | (No) | | | | | | | | | | | | |
| | Yes | 0.46 | 0.16–1.33 | .152 | 0.46 | 0.16–1.31 | .145 | 0.47 | 0.16–1.34 | .157 | 0.50 | 0.17–1.46 | .200 |
| | Fights back | 0.49 | 0.23–1.02 | .056 | 0.48 | 0.22–1.01 | .055 | 0.45 | 0.21–0.94 | .035* | 0.44 | 0.21–0.93 | .031* |
| $R^2$ Hosmer & Lemeshow | | .11 | | | .11 | | | .10 | | | .11 | | |
| $R^2$ Cox & Snell | | .13 | | | .14 | | | .12 | | | .13 | | |
| $R^2$ Nagelkerke | | .18 | | | .19 | | | .17 | | | .18 | | |

[a] AOR, adjusted odds ratio; [b] C.I, Confidence Interval; [c] *p < .05. **p < .01. ***p < .001; [d] $R^2$, Pseudo $R^2$.

behaviour remained significant. HIV-positive women on ART who reported ever being exposed to physical IPV (AOR 0.57, *CI95*: 0.34–0.94, *p* = .028), sexual IPV (AOR 0.50, *CI95*: 0.31–0.82, *p* = .005), or controlling behaviour (AOR 0.56, *CI95*: 0.34–0.94, *p* = .027) were less likely to report ART adherence levels of 95% and over (Table 2). Tertiary education is positively correlated with the odds of achieving optimal ART adherence, except when analysed alongside exposure to emotional IPV (AOR 4.50, *CI95*: 0.99–21.80, *p* = .056). Women who knew their partners were also HIV-positive were more likely to report optimal ART adherence. This positive effect remains significant when analysed alongside exposure to physical IPV

(AOR 1.70, *CI95*: 1.03–2.81, *p* = .040) and sexual IPV (AOR 1.69, *CI95*: 1.02–2.80, *p* = .040); this effect is insignificant when adjusted for controlling behaviour (AOR 1.60, *CI95*: 0.98–2.63, *p* = .062).

The analysis including only women who reported experiencing IPV in the last 12 months revealed similar results (S2 Table). All forms of IPV significantly correlated to suboptimal ART adherence, even when regressed alongside other covariates; physical IPV (AOR 0.58, *CI95*: 0.34–0.98, *p* = .044), sexual IPV (AOR 0.52, *CI95*: 0.30–0.88, *p* = .016), emotional IPV (AOR 0.53, *CI95*: 0.31–0.90, *p* = .021), and controlling behaviour (AOR 0.56, *CI95*: 0.33–0.93, *p* = .026). Higher levels of education and having an HIV-positive partner (when adjusted for physical and sexual IPV) also significantly increased the odds of reporting optimal ART adherence among this subgroup.

## Discussion

In our cross-sectional survey among a broad group of HIV-positive women on ART attending different types of free HIV clinics in government facilities, we examine the prevalence of IPV within the current relationship and its correlation to optimal and suboptimal ART adherence. Our findings reveal a high prevalence of IPV by the current intimate partner and a negative association between exposure to physical IPV, sexual IPV, or controlling behaviour, and odds of achieving optimal ART adherence.

### Prevalence of IPV in the current relationship

Our analyses showed a high prevalence of IPV by the current intimate partner, at 76% (*CI95* = 72–80%). This is significantly higher than the national average (47%) [47]. This was expected, since existing literature indicates that, globally, the lifetime prevalence of IPV among HIV-positive women is high [3,7,11] and greater than that of the general population [1,5,11,12,62]. Our results mirror those of other East African studies [9,11], which also established emotional IPV as the most-reported form of (lifetime and recent) IPV among HIV-positive women. This is followed by physical IPV and sexual IPV. Our study also measured the prevalence of controlling behaviour (39%, *CI95* = 34–44%), which few previous studies have done, despite the small but growing evidence of its adverse health effects [17,63,64]. Although we had anticipated that some women would hesitate to participate in the survey due to the sensitive nature of the topic, all of the healthcare providers reported that the women willingly and openly discussed IPV. This is perhaps because IPV is generally considered a normal, acceptable occurrence in many communities [12,65]. Additionally, IPV is discussed regarding treatment challenges and was not a new topic to most of the participants or healthcare providers.

### ART adherence

Overall, the ART adherence level was high among the women surveyed, with 75% reporting adherence rates of 90% and above. This may result from the women giving socially desirable answers. However, since the survey was done immediately after the routine clinical check-up, which includes a review of the viral load and discussions regarding challenges with medication, we believe there was an increased likelihood that the women gave mostly honest responses. Therefore, the high adherence rates may be credited to the provision of free comprehensive ART and to the vigorous clinic- and community-based patient monitoring and follow-up strategies available. Moreover, the clinics are distributed to reduce travel time and costs. Without these accommodations, patient access and retention in care could be negatively affected [19,66,67]. The high ART adherence rates could also be connected to the high disclosure rate among this group of women. Of the women we surveyed, 94% (n = 380) had disclosed their

status to their current partner. According to Tam et al. [68], on average, approximately 63% of pregnant and post-partum HIV-positive women in sub-Saharan Africa disclose their status to their partners. The higher disclosure rate among our study participants may be due to the broader nature of our sample. It could also be attributed to interventions such as free professional counselling, couple counselling, peer support groups, and social and legal support, which are available at the clinics to help clients address factors that may affect their ART adherence. Non-disclosure is a known barrier to optimal ART adherence because it can result in hesitancy in attending clinic appointments, concealment of ARVs, fear, and a lack of social support [19,68,69]. Moreover, in her study on IPV among HIV-positive women, Hampanda [7] reported that disclosure of HIV-positive status to a partner was associated with increased medication adherence. The low number of women who had not disclosed their status among our participants may account for why the 'disclosure' variable was statistically insignificant, according to the bivariate analysis against ART adherence. Therefore, we did not include it in the multiple logistic regression to avoid inflating the model.

## ART adherence and IPV

Considering an optimal adherence level of ≥95% to analyse the association between exposure to IPV and ART adherence while controlling for other factors, we identified IPV as an independent risk factor for suboptimal ART adherence. Experiencing physical IPV (AOR 0.57, *CI95*: 0.343–0.938, *p* = .028), sexual IPV (AOR 0.499, *CI95*: 0.306–0.810, *p* = .005), or controlling behaviour (AOR 0.563, *CI95*: 0.337–0.936, *p* = .027) from a current intimate partner reduced the odds of an HIV-positive woman reporting optimal ART adherence rates. This finding corroborates studies that link IPV to poor adherence to ARV therapy among women [1,7,70]. According to the healthcare providers in our study, lower adherence among this group of women can be partly attributed to the fact that, during some IPV encounters, some women are chased or forced to flee their homes. In this stressful situation, they may forget to carry their medication; their newfound refuge may make it difficult to maintain their medication schedule or clinic appointments. Moreover, research on IPV reveals that experiencing IPV can lead to psychological distress, depression, and poor mental health [7,28,37,41,62,63] which are predictors of lower ART adherence rates. This could also be an explanation behind the impact IPV has on ART adherence [1,28].

Our findings contrast with those of some studies, such as Gichane et al. [34] and Wilson et al. [35], who found no significant association between IPV and ART adherence among various groups of women in sub-Saharan Africa. The difference in findings may result from several factors. Wilson et al.'s [71] study on women engaged in transactional sex work found no evidence that exposure to IPV was associated with suboptimal (<80%) self-rated ART adherence. One explanation for the difference in findings is that, while we engaged a broader random sample of women who could have a high or low risk of experiencing IPV, Wilson et al. [35] focused on women involved in transactional sex work; this subgroup experiences high rates of overall violence and IPV. These different findings may also result from differences in the recruitment process or IPV measurement, since they annually assessed exposure to IPV in the previous 12 months by a current or former emotional partner while we investigated relationship-specific IPV. Additionally, Wilson et al. noted that their participants were receiving socioeconomic benefits such as transport reimbursement during regular clinic visits, which could have motivated them to be more adherent. Our research differs from Gichane et al.'s [34] study on pregnant and postpartum HIV-positive women in terms of the sample population and measurement. While we used a one-time recording of self-reported adherence at a ≥95% cut-off, Gichane et al. used cumulative ART adherence based on an optimal adherence

cut-off of 100% and recorded over multiple clinic visits. Their study is comparable to that of Hampada et al. [14] in terms of sample population and study design. However, the latter found IPV to be associated with reduced odds of adherence among pregnant and postpartum HIV-positive women.

## Other covariates of ART adherence

Our results indicate that having a tertiary education positively correlates with optimal ART adherence, even when adjusted for IPV exposure and other factors. This aligns with previous research findings showing higher adherence rates among HIV-positive women with higher education compared to their counterparts with little or no education [19,72]. However, it should be noted that the tertiary education subgroup had only 49 respondents, which led to small or no cell numbers for some predictor combinations. Having an HIV-positive partner positively correlated with higher optimal ART adherence rates. This effect remained significant even when adjusted for other factors and exposure to physical IPV and sexual IPV. However, this positive association with optimal ART adherence does not hold when the woman experiences controlling behaviour from the intimate partner. Likewise, women who fought back during IPV encounters had reduced odds of achieving optimal adherence when adjusted for exposure to controlling behaviour, but not when adjusted for physical IPV or sexual IPV. Violence by women during IPV encounters is not uncommon and is known to stem from the need for self-defence, protection of the children, or from fear of severe injury [73–75]. Our findings suggest that the act of fighting back has no effect on ART adherence except in an environment in which controlling behaviour and emotional IPV occurs. We interpret this as being supportive of studies that emphasise the importance of considering controlling behaviour as an independent form of IPV, because it may exert additional effects apart from the other forms of IPV and may also be psychologically more detrimental than previously estimated [17,64]. Since many previous studies had focused on HIV-positive women in PMTCT programs, we examined whether expectant women in our sample would stand out from the group. Pregnancy as an independent variable did not prove significant when we conducted both bivariate analysis against ART adherence and multiple logistic regression. However, this could stem from the low number of pregnant participants *(n* = 14). Based on the latest Demographic Health Statistics, we expected about 6% (*n* = >24) of the women to be pregnant [47].

## Limitations

We recognise several limitations to our research. First, the VAS (like other self-reporting tools) is subject to recall bias (which may cause overestimation) and is prone to eliciting socially desirable responses. Nevertheless, there is sufficient evidence demonstrating that self-report tools correlate with other objective measurement tools, are associated with plasma drug levels, and are useful for identifying vulnerable patients [53,54,76]. Additionally, the questionnaire was administered by the participants' regular healthcare providers, with whom they were familiar and had possibly built a trusted relationship. As mentioned previously, the survey was conducted after the routine clinical check-up, which includes a review of the viral load and ART adherence rate, and discussions on treatment challenges and general wellbeing. We believe that the system for administering the survey increased the likelihood of attaining honest responses from the participants.

Second, we acknowledge that using the ≥95% cut-off to represent optimal ART adherence is controversial because it is based on obsolete therapy; advanced agent combinations can now effectively suppress the HIV viral load at lower ART adherence levels. Therefore, we reran the analysis using a lower cut-off (≥85%) and found that the results converge, with the only

difference being that women whose partners 'sometimes got drunk' or who were 'sometimes afraid of their partners' also had lower odds of achieving ART adherence at the $\geq$85% cut-off. We are, therefore, confident that our findings on the effects of IPV are robust. The decision to measure exposure to IPV within the current intimate relationship (regardless of when it occurred) and compare it to 30-day self-reported ART adherence could limit our ability to associate IPV with suboptimal ART adherence. However, our interest was investigating whether past or ongoing IPV within the relationship impacted ART adherence. Our rationale is that specific IPV incidents may not affect medication adherence, but living in an environment in which IPV occurs does influence a woman's ability to maintain optimal ART adherence. This is further supported by the fact that the second analysis, which we ran including only the women who experienced IPV in the last 12 months, revealed similar results to the relationship-specific analysis.

Due to resource limitations, we were unable to expand our capacity to capture cases of defaulters whose adherence (to clinic appointments or medication) reduced over time until they ultimately dropped out of the program. These are perhaps the most vulnerable group and future research should consider tracking them down to establish their reason for attrition. Additionally, the cross-sectional nature of the study means that no temporality can be inferred; therefore, we cannot make any causality inferences. Nevertheless, we believe our findings are useful, as they contribute evidence for an association between IPV and undesirable clinical outcomes. We also offer insight into where strategies can be introduced to improve ART adherence outcomes.

On the other hand, our study has several strengths. First, we randomly recruited women of diverse ages (18–69 years) receiving ARV treatment in different types of free HIV clinics at government health facilities. This indicates that our sample accounted for the varying characteristics of women receiving ART in Kenya. Previous studies have tended to focus on key populations (and rightly so), participants of ongoing trials, or couples jointly participating in interventions, which can limit the generalisability of results.

## Conclusion and future practices

Despite the great advances in ART such as increased drug efficacy, improved access to ARVs, and convenient regimens, achieving optimal ART adherence remains a challenge among some groups. As Shubber et al [67] noted, there is no single intervention that will be sufficient to ensure high levels of ART adherence. What is needed is a combination of interventions that aim at strengthening aspects or undermining barriers which are linked to adherence. Similar to some existing studies, our findings show a high prevalence of IPV among HIV-positive women, less than optimal ART adherence rates, and a significant negative association between IPV and optimal ART adherence. Therefore, basic screening for exposure to IPV by healthcare providers could help in early identification, monitoring, and extra support for exposed women since we believe they are at risk of poor ART adherence. Moreover, since it is already established that IPV has negative effects on women's health [22,64,77,78], any reduction of the harmful effects would generally improve the HIV-positive women's quality of life. The IPV screening could be integrated at the earlier stages of ART initiation. Alternatively, it could be done at the healthcare provider's discretion, based on the woman's responsiveness to the therapy. The clinical officers and nurses involved in this study reported that participants willingly discussed their IPV experience. The former also expressed interest in receiving training on IPV. A review on programs in sub-Saharan Africa which have implemented IPV screening and counselling services in healthcare settings, reported that the interventions were positively received by both healthcare providers and clients [2]. Healthcare workers could therefore be navigation points for women exposed to IPV who are willing to be assisted.

Nevertheless, there is need for future research to investigate further how IPV affects ART adherence. One recommendation would be to focus the studies on women with poor ART adherence and also tracking down women who drop out of ART programs. Furthermore, controlling behaviour remains an underestimated and thus understudied form of IPV despite growing evidence of its adverse effects. We encourage future researchers to consider including it in their investigations.

## Supporting information

**S1 Table. Propotional stratified sampling based on the number of women on ART and location of clinic.**
(DOCX)

**S2 Table. Multiple regression models of factors associated with ART adherence among women who experienced IPV in the last 12 months.**
(DOCX)

**S3 Table. Multiple regression models of factors associated with ART adherence including clinics as a predictor variable.**
(DOCX)

## Acknowledgments

We would like to thank the Ministry of Health/AMPATH clients for agreeing to participate in this survey. We extend our gratitude to the healthcare providers for administering the questionnaires and interacting with the women. We thank the Ministry of Health/AMPATH leadership for making this research possible.

## Author Contributions

**Conceptualization:** Bornice C. Biomndo, Nils Lahmann, Lukoye Atwoli.

**Data curation:** Bornice C. Biomndo.

**Formal analysis:** Bornice C. Biomndo, Alexander Bergmann, Lukoye Atwoli.

**Funding acquisition:** Bornice C. Biomndo.

**Investigation:** Bornice C. Biomndo, Alexander Bergmann, Lukoye Atwoli.

**Methodology:** Bornice C. Biomndo, Alexander Bergmann, Nils Lahmann, Lukoye Atwoli.

**Project administration:** Bornice C. Biomndo, Nils Lahmann, Lukoye Atwoli.

**Resources:** Bornice C. Biomndo, Nils Lahmann, Lukoye Atwoli.

**Software:** Bornice C. Biomndo, Alexander Bergmann.

**Supervision:** Bornice C. Biomndo, Nils Lahmann, Lukoye Atwoli.

**Validation:** Bornice C. Biomndo, Alexander Bergmann, Lukoye Atwoli.

**Visualization:** Bornice C. Biomndo, Alexander Bergmann, Lukoye Atwoli.

**Writing – original draft:** Bornice C. Biomndo, Nils Lahmann.

**Writing – review & editing:** Bornice C. Biomndo, Alexander Bergmann, Lukoye Atwoli.

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
