## [Decision Letter · Decision Letter 0]

16 Sep 2020

PONE-D-20-21450

Intimate partner violence a barrier to antiretroviral therapy adherence among HIV positive women: Evidence from government facilities in Kenya

PLOS ONE

Dear Dr. Biomndo,

Thank you for submitting your manuscript to PLOS ONE. After careful consideration, we feel that it has merit but does not fully meet PLOS ONE’s publication criteria as it currently stands. Therefore, we invite you to submit a revised version of the manuscript that addresses the points raised during the review process.

Thank you for your submission. This manuscript has the potential to be a timely contribution to the literature. However, it needs substantial revisions to improve clarity and complete presentation of methods and results. The reviewers have both provided very detailed comments on each section of the manuscript. Below I am providing some more large-scale areas that should be addressed before resubmission:

1) There are numerous grammatical errors and missing words throughout the manuscript that make it difficult to understand what the authors are saying at times. Please consider enlisting the help of a copy editor prior to resubmission.

2) Please ensure consistency of terminology, specifically related to the key concepts in the manuscript (e.g., intimate partner violence (IPV) including physical IPV, sexual IPV, and emotional IPV; antiretroviral therapy). Define terms that may not be well understood (e.g., mixed clinics). Abbreviations should be introduced and then used consistently.

3) Citations need to be consistent per PLOS One format (Vancouver Style) and should be used for each claim that references supporting literature. For example, the introduction cites "IPV is a known risk factor for HIV infection..." without a citation. Also, all tables/figures should be cited in the text.

4) The methods section needs additional details about measures and procedures including: (a) How were the 12 clinics selected (e.g., was it a random sample)?; (b) For inclusion criteria, what were the parameters around "in an intimate relationship"? Did this have to be at the time of the study? At any point in their life?; (c) Were the IPV measurements for lifetime? If so, why was this time frame used when ART adherence focused on the last 30 days?; (d) Is the "recruiter" a trained study staff member or their healthcare provider? Did they have resources available for someone who was actively experiencing IPV and asked for resources? Were participants excluded if there was a safety concern? Were the interviews conducted in private? Were they conducted with paper surveys? Was there an incentive?; (e) How many participants declined participation? What was the reasoning?

5) The results section should be streamlined. First of all, a descriptive/demographic table of the sample is needed. Second, the current tables contain a large amount of information. The authors should consider how to condense this information in an easily digestible way.

6) As noted by Reviewer #1, the conclusion could benefit from a future research or future directions section.

We look forward to receiving your revised manuscript.

We look forward to receiving your revised manuscript.

Kind regards,

Michelle L. Munro-Kramer, PhD, CNM, FNP-BC

Academic Editor

PLOS ONE

2. Please state whether you validated the questionnaire prior to testing on study participants.

Please provide details regarding the validation group within the methods section.

Reviewers' comments:

Reviewer's Responses to Questions

**Comments to the Author**

1. Is the manuscript technically sound, and do the data support the conclusions?

Reviewer #1: Yes

Reviewer #2: Yes

2. Has the statistical analysis been performed appropriately and rigorously? 

Reviewer #1: No

Reviewer #2: Yes

3. Have the authors made all data underlying the findings in their manuscript fully available?

Reviewer #1: Yes

Reviewer #2: Yes

4. Is the manuscript presented in an intelligible fashion and written in standard English?

Reviewer #1: No

Reviewer #2: No

5. Review Comments to the Author

Reviewer #1: This manuscript has the potential to contribute important information to the literature on women’s ART adherence and IPV. The authors have a large sample size and the findings have extremely important implications on future practices. Several things that should be addressed before this manuscript is ready for publication.

General: Overall, there are many grammatical errors and words that appear to be missing (including in the title). The authors should considering hiring a copy editor to carefully review the manuscript. The authors should also ensure all citations are in a consistent format (e.g., pg. 14 line 241).

Abstract: In the results of the abstract the authors state protective factors but it is unclear if they are protective factors regarding increased ART adherence or IPV.

Introduction:

Overall: The authors state a “gap in the literature” but then cite several studies. It seems that the authors should report conflicting results in the literature rather than a gap. Given the specific populations previous studies focused on, this study adds to the literature by recruiting from multiple types of clinics across locations. I believe this should be further emphasized in the introduction to provide additional rationale of the importance of this study.

Specific:

*The authors state that most studies are conducted in specific regions/cities in Africa, which suggested that the present study would be conducted across sub-Saharan Africa. Discussion on the limitations of location-specific studies should be limited since this study is conducted in Kenya and not across sub-Saharan Africa.

Methods:

Overall: I have some concerns and questions regarding the measures and data analysis.

Specific:

*If IPV influences ART adherence, why were women included only if they had been on ART for at least 6 months? Is it possible that women may not be on ART in the past 6 months because of IPV?

*Adherence was measured in the past 30 days, yet from the description, it appeared that IPV was a lifetime measure. The authors should explain why IPV was not measured in the same timeframe as adherence (or at least more recent—past 3 months).

*The authors should define “mixed clinic” as this is unclear.

*Were participants provided any compensation for participating?

*Was there any missing data? If so, how was it accounted for during analyses? Analyses that included multiple locations often use a nested model to account for differences. Did the authors include clinics in analyses?

Results:

Overall: The tables are somewhat difficult to follow and incorporate a lot of information. Since there are 4 very large tables, the authors should consider a more succinct manor in presenting results.

Specific:

*Figure 1 is not mentioned in the manuscript and it should be referred to as a table rather than a figure.

*A demographics table would be helpful, especially if the authors minimize or combine tables 1 – 4.

*It is unclear what ~ refers to in the tables.

*Many of the variables listed in the tables are not described in the Method/Measures section.

Discussion:

Overall: These results have very important implications for future research and practices. The authors appear to mention these briefly in the Conclusion section. It would be beneficial to provide a “Future Research” or “Future Practices” section in the Discussion that discusses next steps. The Conclusion should briefly summarize the study results and implications.

Reviewer #2: After careful consideration and review of manuscript number PONE-D-20-21450 entitled, “Intimate partner violence a barrier to antiretroviral therapy adherence among HIV positive women: Evidence from government facilities in Kenya”, it is my recommendation for the authors to submit a major revision of the paper. Details are provided below:

Abstract

1. The introduction section is confusing because the authors note that studies on key populations have produced divergent evidence regarding “this association” but it is unclear what “this association” references…intimate partner violence and poor medication adherence or intimate partner violence and virological failure?

2. It’s unclear what “mixed clinics” means.

3. The study design and procedures are unclear in the methods section. I recommend conveying the analysis in a brief, succinct manner to allow room for a sentence or two on the study design and procedures.

4. The study implications are vague. There must be more recommendations than merely integrating IPV screening and counseling services within HIV clinics.

Introduction

1. Although there are no overall IPV prevalence estimates available from sub-Saharan Africa, there have been numerous studies conducted from which prevalence estimates can be cited, within a range.

2. It is inappropriate to claim that IPV is “recognized as a factor behind low uptake and engagement of HIV care” with only one citation provided.

3. There are sentences written without any citations which is unacceptable. For instance on pg. 5, lines 58-60, multiple citations are needed because that sentence references “studies”.

4. Typos and grammatical errors are present and should be corrected.

5. Pg. 5, lines 81-83 presents a sentence that is awkwardly worded; it is difficult to understand the purpose and meaning of the sentence.

6. In one area, “partner violence” is used and in other areas “intimate partner violence” is used. Is this supposed to represent a distinction?

7. The claim is made that “there needs to be an expansive approach that encompasses the diverse population of HIV positive women” but it is unclear what this approach actually is and how this approach serves as the foundation of the paper.

Methods

8. For consistency, once ART has been defined, it is necessary to utilize the acronym throughout the paper. In the methods section, this was not the case.

9. The eligibility criteria does not include participants self-reporting as women; was this not an inclusion criteria?

10. The national IPV prevalence rate was mentioned in the methods section but not included in the introduction section.

11. The term, intimate partner violence or IPV, is loosely used and not consistently used. In some areas the word, “violence” is used. Consistency is necessary. Specifically, “physical violence” should be “physical IPV”, etc.

12. It is unclear what is meant by “mixed clinics”.

13. More details are needed regarding the administration of the survey (e.g., was the location in a private office, computer administered, etc.).

14. How was participation handled for women who were at the clinic with a partner?

15. It is inappropriate to consider the variables in the analysis as “predictors” because the data is cross-sectional in nature, not longitudinal.

Results

16. The traditional “Table 1” is missing. This table typically provides descriptive statistics to describe the sample.

17. Figure 1 is included but never referred to in the text of the manuscript.

18. Again, in stating the different types of violence, it should be specified that these types of violence are IPV (emotional IPV, sexual IPV).

19. The table titles are misleading and need to account for both the primary independent variable and dependent variable.

Discussion

20. The authors need to be clear that the discussion points are centered around “lifetime” IPV rather than just stating “prevalence of IPV” – this should be “prevalence of lifetime IPV”.

21. The authors make an incorrect declarative statement (pg. 15, line 255). The study findings can only corroborate studies that document an association between IPV and poor adherence to ARV medication, not the association between IPV and poor uptake of ARV medication.

22. Pg. 16, line 287-290 – This sentence is grammatically incorrect.

23. The implications of the study findings are general and do not provide significant details on how the finding could inform practice and policy. Additionally, there is lack of specificity in terms of future research directions resulting from the current study findings.

24. Limitations should be discussed regarding the time anchor for the measure, lifetime IPV. Recent IPV may have been a better indicator as it relates to impeding ART adherence.

6. PLOS authors have the option to publish the peer review history of their article (what does this mean?). If published, this will include your full peer review and any attached files.

Reviewer #1: No

Reviewer #2: No

---

## [Author Response · Author response to Decision Letter 0]

31 Oct 2020

Editor’s comments

1) There are numerous grammatical errors and missing words throughout the manuscript that make it 

difficult to understand what the authors are saying at times. Please consider enlisting the help of a copy editor prior to resubmission.

Thank you for the feedback. Reviewers #1 and #2 also commented on the grammatical errors and typos. We have therefore changed the title to read ‘Intimate partner violence is a barrier to antiretroviral therapy adherence among HIV-positive women: Evidence from government facilities in Kenya’. Furthermore, we have rectified the typos and grammatical errors within the manuscript and hired a professional copy editor to review the final draft.

2) Please ensure consistency of terminology, specifically related to the key concepts in the manuscript 

 (e.g., intimate partner violence (IPV) including physical IPV, sexual IPV, and emotional IPV; antiretroviral therapy). Define terms that may not be well understood (e.g., mixed clinics). Abbreviations should be introduced and then used consistently. 

We thank you for this correction. Reviewers #1 and #2 also commented that the term ‘mixed clinics’ was unclear. Reviewer #2 pointed out that the term ‘IPV’ was loosely and inconsistently used, that consistency was necessary, and that the different types of violence should be specified as physical IPV, emotional IPV, etc. We have revised the manuscript to clearly and consistently identify the forms of violence as IPV and to consistently use the term IPV. We have rewritten the part of the abstract that mentioned ‘mixed clinics’ (Pg. 2 line 28). Within the methods section, we have clarified in Pg. 8 lines 145 and 146 that ‘mixed clinics refer to non-specialized clinics where female and male HIV-positive adults are reviewed and given medication refills.

3) Citations need to be consistent per PLOS One format (Vancouver Style) and should be used for 

each claim that references supporting literature. For example, the introduction cites "IPV is a known risk factor for HIV infection..." without a citation. Also, all tables/figures should be cited in the text. 

Thank you for pointing out this oversight. Reviewers #1 and #2 also commented on citation. Specifically, in the previous manuscript, Reviewer #1 noted that all citations should be in a consistent format and gave the example of Pg. 14 line 241, which was not consistent with the format. Reviewer #2 pointed out that it was inappropriate to claim that IPV is “recognised as a factor behind low uptake and engagement of HIV care” and provide only one citation. Additionally, some sentences (e.g. Pg. 5 lines 58-60) were missing citations. In the revised manuscript we have ensured that all references are appropriately cited (i.e. Pg. 15 line 259). We have added six citations to support the introductory statement that ‘Intimate Partner Violence (IPV) is recognised as a factor behind low uptake and engagement of HIV management services’, and added citations in Pg. 4 lines 58 and 59 for the studies we referred to. Moreover, regarding the claim that ‘IPV is a risk factor for HIV infection among women....’ (Pg. 4 line 59), we have added two citations and separately referenced lines 60-62, which we had previously erroneously combined into one citation. 

For the tables and figures, we have included the traditional Table 1 as advised and created a new table that we believe gives a concise and clear summary of the logistic regression analysis results. We have also confirmed that all relevant tables are mentioned in the texts. 

4) The methods section needs additional details about measures and procedures including: 

(a) How were the 12 clinics selected (e.g., was it a random sample)?

Within the study design and setting section, we have added a sentence explaining that the 12 clinics were selected based on the number of active HIV-positive women on ART they, and also to represent the different counties (regions) in western Kenya. Consequently, we included large clinics with high numbers of women on ART in the different counties. Additionally, smaller clinics were selected to have representation from less populous communities. We have rewritten the information in the methods section (Pg.5 lines 93–97) to clarify this.

(b) For inclusion criteria, what were the parameters around "in an intimate relationship"? Did this have to be at the time of the study? At any point in their life? 

(c) Were the IPV measurements for lifetime? If so, why was this time frame used when ART adherence focused on the last 30 days? 

We appreciate the detailed feedback and we acknowledge that the study’s definition and measurement of IPV was somewhat unclear in the previous manuscript. Reviewer 1# also commented that, while adherence was measured over the past 30 days, the description suggested that IPV was a lifetime measure. Reviewer #1 stated that we should explain why IPV was not measured in the same timeframe as adherence (or at least more recently, such as the past 3 months). Since these comments are related, we decided to combine them and offer a comprehensive response.

Our study focuses on the current relationship and the occurrence of IPV within the relationship’s lifetime. We have clarified this by including within the data collection section the description of the IPV measurement and a sentence defining an intimate partner as someone to whom the woman was currently married (whether in a monogamous or polygamous marriage) or a person with whom she was in a romantic relationship (Pg. 6 Lines 118 and 119). We have also included the words ‘current intimate partner’ and ‘current relationship’ in the abstract, results, and discussion sections whenever we refer to the partner or relationship. 

We chose to use a relationship-specific timeframe for IPV exposure and ART adherence in the 30 days prior to the survey, because our interest was investigating whether living in an environment where IPV occurs (whether past or ongoing) affects a woman’s ability to adhere to treatment. We hypothesise that there is a potential link. Our reasoning is that, although a specific IPV incident may not affect medication adherence, the impact of living in such a setting is what influences the woman’s ability to maintain optimal ART adherence.

(d) Is the "recruiter" a trained study staff member or their healthcare provider? Did they have resources available for someone who was actively experiencing IPV and asked for resources? Were participants excluded if there was a safety concern? Were the interviews conducted in private? Were they conducted with paper surveys? Was there an incentive?

Reviewer #1 also asked whether the women were offered any compensation for participating. We have added more information on the recruiters, the survey administration process, and the compensation within the procedure section. The ‘recruiters’ were the women’s regular healthcare providers – i.e. clinical officers and nurses who provide and supervise ART at the clinics. The healthcare providers were trained on the aims of the study, the recruitment process, and questionnaire administration. A list of in-house resources (social workers and, where available, a legal aid office) and external resources (government authorities, non-governmental organisations in the area that work in gender-based violence prevention/women’s empowerment) was compiled and offered to women who reported exposure to IPV.

The survey was conducted right after the woman’s routine clinical check-up; therefore, it took place in the same examination room. Although this was very convenient and practical, unfortunately, it meant that if the examination room was shared, the healthcare providers had to ensure some form of privacy when administering the questionnaire. They did so either by asking their fellow healthcare provider to temporarily leave the room or by creating physical distance by moving their desks or dividing the room with a curtain. If the woman was accompanied by another adult or a child who was old enough to understand the conversation, the person/child was politely asked to leave the room before the questionnaire was administered. Women who participated in the survey were compensated for their time and effort in form of cash, which was referred to as ‘transport money’. This was only mentioned and given after the woman completed the questionnaire. This information has been added within the procedure section.

In the data collection section, we have added a sentence clarifying that the questionnaire was in paper form (Pg. 7 lines 142 and 143).

(e) How many participants declined participation? What was the reasoning?

 Thank you for raising this question. In the discussion section, we have added information stating that, despite anticipating some hesitance from the women, all of the healthcare providers reported that the women they approached willingly and openly discussed their IPV experience. We have also included the reasons that we think the women were receptive to the survey (Pg. 14 Lines 244 -248). Although some healthcare providers did require more time to recruit women, this was due to not having enough women who fit the criteria, rather than the women declining to participate.

5) The results section should be streamlined. First of all, a descriptive/demographic table of the sample 

is needed. Second, the current tables contain a large amount of information. The authors should consider how to condense this information in an easily digestible way. 

Thank you for this feedback. We have added Table 1, which contains descriptive information about the sample. The previous tables were certainly large and information-heavy. Therefore, we have developed a single table (Table 2) that combines the previous four models in a concise way that is easier to follow. We decided not to report the results of the single regression analyses within an extra table; instead, we focus on the full regression models, which are more informative.

Within Table 2, Pseudo R2 values are reported for transparency even though the aim of our model is to explore the association between the independent variables and the dependent variable, and not to predict the dependent variable [1, 2]. Nevertheless, the arguably low Pseudo R2 values were not surprising to us since we expected that there are other variables that affect ART that were not measured in our survey [3].

6) As noted by Reviewer #1, the conclusion could benefit from a future research or future directions 

 section.

Reviewer #1 commented that it would be beneficial to provide a ‘Future Research’ or ‘Future Practices’ section that discusses next steps. The Conclusion should briefly summarise the study results and implications. We agreed that the discussion and conclusion could be improved and have rewritten them to expound more on the implications of our study; we have specified the practical steps that can be taken and suggested future research. Specifically, we have added a ‘Future practices’ segment within the discussion section (Pg. 18 line 351) that expounds on the practicality of integrating basic IPV screening and counselling within HIV clinics and the role that healthcare providers can play in this regard. We also indicate the need for strategies at the societal level to create awareness on the negative effects of IPV, the laws that protect against it, and the existing resources that are available for women exposed to IPV. To conclude, we have expressed the need for more evidence on the direct and indirect benefits of integrating IPV screening and counselling for women in healthcare settings. We have called on future researchers to target vulnerable groups (e.g. women who drop out of ART programs) and to produce more evidence on the negative effects of controlling behaviour.

Reviewer #1

Abstract

In the results of the abstract the authors state protective factors but it is unclear if they are protective factors regarding increased ART adherence or IPV.

Noted. We have rewritten the sentence to make it clear that the protective factors regard increased ART adherence (Pg. 2 line 39).

Introduction:

Overall: The authors state a “gap in the literature” but then cite several studies. It seems that the authors should report conflicting results in the literature rather than a gap. Given the specific populations previous studies focused on, this study adds to the literature by recruiting from multiple types of clinics across locations. I believe this should be further emphasized in the introduction to provide additional rationale of the importance of this study.

This is true. Our intention was to emphasise that the studies that have aimed to fill this gap in the literature by focusing on specific subsets of women have produced contrasting and inconclusive evidence. Therefore, our study aims to contribute by adding evidence from a broad group of women. We now realise that area was not well-written in the previous manuscript. We have revised parts of the introduction section to make this point more clearly (Pg. 5 lines 74-82). 

Specific:

*The authors state that most studies are conducted in specific regions/cities in Africa, which suggested that the present study would be conducted across sub-Saharan Africa. Discussion on the limitations of location-specific studies should be limited since this study is conducted in Kenya and not across sub-Saharan Africa.

By mentioning specific locations in our paper, our intention was to highlight that there are studies from other sub-Saharan African countries which may be comparable to our setting and population in Kenya. We did not intend to imply that the study location could be a limiting factor or that our study was conducted in several locations. Any discussion or comparison to a specific study has focused on the sample population (e.g. pregnant and postpartum women, transactional sex workers) or the method of measuring ART adherence (cumulative ART adherence and cut-off points). However, we now see how highlighting specific locations could have a different effect than we anticipated; therefore, we have limited this in the revised manuscript.

Methods:

Overall: I have some concerns and questions regarding the measures and data analysis.

Specific:

*If IPV influences ART adherence, why were women included only if they had been on ART for at least 6 months? Is it possible that women may not be on ART in the past 6 months because of IPV?

We realized that the wording we used to explain this criteria was misleading, and we have rewritten it in Pg.6 lines 100–103. The six-month limit refers to the length of time since initiation of ART – not to the length of time that the women had actively been on ART prior to the survey. Therefore, it did not matter whether or for how long the woman had been off ART prior to the study. They were included as long as the onset of ART was more than six months prior to the survey. Our outcome of interest was ART adherence. Because we hypothesise that some factors affecting adherence are related to the initiation stages of ART, we tried to minimise these factors by capturing women who may have been accustomed to the ART routine.

*Was there any missing data? If so, how was it accounted for during analyses? Analyses that included multiple locations often use a nested model to account for differences. Did the authors include clinics in analyses?

We have added a brief paragraph in the data analysis section ( from Pg. 9 line 182) explaining that a missing value analysis was performed and none of the variables had more than 5% missing values; we also did not find any systematic relationships between the missing values. Regarding the nested structure of our data, we have added on Pg. 9 line 190- 193 that we decided against a multilevel modelling procedure because, with as few as 12 clusters, fixed-effect estimates associated with level-2 predictors could have been severely biased.

Results:

Overall: The tables are somewhat difficult to follow and incorporate a lot of information. Since there are 4 very large tables, the authors should consider a more succinct manor in presenting results.

Specific:

*Figure 1 is not mentioned in the manuscript and it should be referred to as a table rather than a figure

*A demographics table would be helpful, especially if the authors minimize or combine tables 1 – 4.

Thank you for this feedback. We have made major revisions to the tables and figures, as suggested. We developed Table 1, which contains demographic information on the participants. We have combined the previous tables (Tables 1–4) into a single table (Table 2). 

 We removed the table containing information on the clinics (previously erroneously identified as Figure 1) and offered it as extra supplementary information, since it does not directly relate to the results and only serves as extra information. Additionally, we deleted Figure 2 since the information that was presented is now available in Table 1 and within the text in the results section.

*It is unclear what ~ refers to in the tables.

Noted. We have deleted the symbol since we realised that the information was clear without it.

*Many of the variables listed in the tables are not described in the Method/Measures section.

We apologise for this oversight. On Pg. 7 line 139–142 we have included all of the variables that are later presented in the tables

Reviewer #2

After careful consideration and review of manuscript number PONE-D-20-21450 entitled, “Intimate partner violence a barrier to antiretroviral therapy adherence among HIV positive women: Evidence from government facilities in Kenya”, it is my recommendation for the authors to submit a major revision of the paper. Details are provided below:

Abstract

1. The introduction section is confusing because the authors note that studies on key populations have produced divergent evidence regarding “this association” but it is unclear what “this association” references…intimate partner violence and poor medication adherence or intimate partner violence and virological failure?

Noted. By ‘this association’ we were referring to IPV’s link to adverse clinical outcomes. However, since our paper focuses on poor ART adherence, we have decided to rewrite the statement to specify the association between IPV and poor ART adherence (Pg. 2 line 23 -25).

2. (This comment has already been responded to under Editor’s comment #2)

3. The study design and procedures are unclear in the methods section. I recommend conveying the analysis in a brief, succinct manner to allow room for a sentence or two on the study design and procedures.

Noted. We have rewritten the section to read ‘We sampled 408 HIV-positive women receiving free ART from different types of HIV clinics at government health facilities, assessing for IPV exposure by a current partner, ART adherence rate, and other factors that affect ART adherence (e.g. education, disclosure). ART adherence rates were measured using the Visual Analogue Scale (VAS); responses were dichotomised at a ≥95% cut-off. Multiple logistic regression models assessed the association between the independent variables and ART adherence.

4. The study implications are vague. There must be more recommendations than merely integrating IPV screening and counselling services within HIV clinics.

Thank you for highlighting this. We have revised the conclusion section to read ‘IPV is associated with suboptimal ART adherence rates among a broad group of HIV-positive women. ART programs should consider incorporating basic IPV interventions into regular clinic services and using healthcare providers as information and navigation points for women exposed to IPV.’ 

Introduction

1. Although there are no overall IPV prevalence estimates available from sub-Saharan Africa, there have been numerous studies conducted from which prevalence estimates can be cited, within a range.

Thank you for this suggestion. We were able to find some estimates, which we have included in Pg. 4 lines 52 and 53.

2. and 3. (These comments on citation have already been addressed under the Editor’s comment #3)

4. (This comment on typos and grammatical errors has been addressed under Editor’s comment #1)

5. Pg. 5, lines 81-83 presents a sentence that is awkwardly worded; it is difficult to understand the purpose and meaning of the sentence.

Noted. We realise our mistake and have rewritten the sentence to read ‘So far, the evidence from sub-Saharan Africa has been based on research among key populations, such as transactional sex workers and women at Prevention of Mother-to-Child Transmission (PMTCT) clinics (Pg. 5 lines 77–79). 

6. In one area, “partner violence” is used and in other areas “intimate partner violence” is used. Is this supposed to represent a distinction?

No, the intent was not to make any distinction. We acknowledge this mistake and have corrected it by consistently mentioning IPV whenever we talk about violence.

7. The claim is made that “there needs to be an expansive approach that encompasses the diverse population of HIV positive women” but it is unclear what this approach actually is and how this approach serves as the foundation of the paper.

Thank you for pointing this out. To clarify what the approach is, we have revised this claim on Pg. 5 line 79–82 to read ‘However, to determine whether IPV exerts an overarching influence on ART adherence, the association between both variables must be explored among a broader sample of HIV-positive women with diverse socioeconomic characteristics who are receiving the available standard ART from different types of HIV clinics.’

Methods

8. For consistency, once ART has been defined, it is necessary to utilize the acronym throughout the paper. In the methods section, this was not the case.

Noted. We have revised the manuscript and ensured that the abbreviation is used consistently throughout.

9. The eligibility criteria does not include participants self-reporting as women; was this not an inclusion criteria?

The women were recruited from patient lists which are developed from the official medical records. The patient lists are a list of the clients who have clinic appointments on that day. The recruiters used the patient lists to randomly select participants who, based on their official medical records which the clients filled in during their first clinic encounter, are categorised as ‘female’. Since this categorisation was predetermined by how the participant self-identifies in their official medical records, we did not see the need to consider it as an extra criteria. Nevertheless, in the questionnaire, we were conscious to use the term ’partner’ when gathering information about the relationship with the intimate partner, and ‘husband/partner’ when measuring exposure to IPV.

10. The national IPV prevalence rate was mentioned in the methods section but not included in the introduction section.

Thank you for highlighting this omission. We have included the national IPV rate in the introduction section (Pg. 4 lines 55 and 56).

11. (This comment on inconsistent use of the term IPV has been addressed under Editor’s comment #2)

12. (This comment on mixed clinics has been addressed under Editor’s comment #2)

13. More details are needed regarding the administration of the survey (e.g., was the location in a private office, computer administered, etc.).

14. How was participation handled for women who were at the clinic with a partner?

Added. We have added in Pg. 7 lines 142 and 143 a sentence explaining that the questionnaire was in paper form. Furthermore, within the procedure section (from Pg.8 line 155), we have included more information on where and how the questionnaire was administered, as well as the process. Specifically, we have explained that the survey was administered in the regular examination room by the healthcare provider immediately after the routine clinic visit was completed. Although very practical, the challenge with this set-up was that the healthcare provider had to ensure maximum possible privacy. We have also stated that, in cases where the woman was accompanied by another person or a child who was old enough to understand the questions, the person or child was politely asked to leave the room. 

15. It is inappropriate to consider the variables in the analysis as “predictors” because the data is cross-sectional in nature, not longitudinal.

Noted. We reviewed the manuscript and have now limited the use of the word ‘predictor’ to only describe the relationship between two variables in the process of statistical modelling. This is in line with OECD’s definition of the term ‘prediction’ within statistical contexts[4]. When inferring to the real world, we have used the term ‘independent variable’. We regret any previous oversight in this regard.

Results

16. The traditional “Table 1” is missing. This table typically provides descriptive statistics to describe the sample.

Added. We have developed Table 1, which contains demographic information for the participants.

17. Figure 1 is included but never referred to in the text of the manuscript.

We removed the table containing information on the clinics (previously erroneously identified as Figure 1) and offered it as extra supplementary information since it does not directly relate to the results and only serves as extra information.

18. (This comment has been responded to under Editor’s comment #2)

19. The table titles are misleading and need to account for both the primary independent variable and dependent variable.

Thank you for highlighting this. We have corrected this in Tables 1 and 2, which we have developed to summarise the results.

Discussion

20. The authors need to be clear that the discussion points are centered around “lifetime” IPV rather than just stating “prevalence of IPV” – this should be “prevalence of lifetime IPV”.

Agreed. We needed to be clearer on the definition of the prevalence of IPV. We have rewritten the subtitle to read ‘Prevalence of IPV within the current relationship’ on Pg. 10 line 199 and Pg.14 line 236.

21. The authors make an incorrect declarative statement (pg. 15, line 255). The study findings can only corroborate studies that document an association between IPV and poor adherence to ARV medication, not the association between IPV and poor uptake of ARV medication.

This is correct and we regret this error. We have corrected the statement to read that the finding corroborates studies that link IPV to poor adherence to ARV therapy among women (Pg. 15 line 276 and 277).

22. Pg. 16, line 287-290 – This sentence is grammatically incorrect.

Noted. After reassessing the sentence, we agreed that deleting it would not alter the point we were aiming to make. We deleted the sentence and took the opportunity to restructure the argument about the effects of controlling behaviour (Pg. 17 line 308- 314).

23. The implications of the study findings are general and do not provide significant details on how the finding could inform practice and policy. Additionally, there is lack of specificity in terms of future research directions resulting from the current study findings.

We agree that the conclusion could be improved and we have added a section on ‘future practices’ within the discussion section (Pg. 18 line 351), which expounds on the implications of our study and specifies the practical steps that can be taken. Additionally, we completely rewrote the conclusion section to include suggestions on future research (Pg. 19 lines 372).

24. Limitations should be discussed regarding the time anchor for the measure, lifetime IPV. Recent IPV may have been a better indicator as it relates to impeding ART adherence.

Thank you for pointing this out. We took the comment into consideration and included in the limitation section (Pg. 18 from line 337) that, although this could have limited our ability to associate IPV to suboptimal ART adherence, the reason for measuring exposure to IPV within the current intimate relationship (regardless of when it occurred) and comparing it to the 30-day self-reported ART adherence was that our aim was to investigate the impact that past or ongoing IPV within the relationship had on ART adherence. Our rationale is that specific IPV incidents may not affect medication adherence; rather, living in an environment in which IPV occurs is what influences the woman’s ability to maintain optimal ART adherence.

 

Journal Requirements

2) Please state whether you validated the questionnaire prior to testing on study participants. Please provide details regarding the validation group within the methods section.

3) We note that you have indicated that data from this study are available upon request. PLOS only allows data to be available upon request if there are legal or ethical restrictions on sharing data publicly.

We have uploaded our data set to Zenodo and it is available under the following DOI: 10.5281/zenodo.4135394.

4) Please include a separate caption for each figure in your manuscript.

We apologise for the oversight. We have included the necessary captions for the tables (we no longer have any figures).

References

1. Forst J. Regression analysis : An intuitive guide for using and interpreting linear models. Pennsylvania, United States: Statistics By Jim Publishing; 2019.

2. Editor MB. The Minitab Blog [Internet]. Minitab, editor. State College2013. [cited 2019]. Available from: https://blog.minitab.com/blog/adventures-in-statistics-2/regression-analysis-how-do-i-interpret-r-squared-and-assess-the-goodness-of-fit.

3. Giselmar A. J. Hemmert LMS, Jan Wieseke, and Heiko Schimmelpfennig. Log-likelihood-based Pseudo-R2 in logistic regression: Deriving sample-sensitive benchmarks. Sociol Method Res. 2018;47(3).

4. Marriott FHC. A Dictionary of Statistical Terms. 5th ed. Harlow, Essex: Longman Scientific & Technical; 1990. 223 p.

---

## [Decision Letter · Decision Letter 1]

11 Jan 2021

PONE-D-20-21450R1

Intimate partner violence is a barrier to antiretroviral therapy adherence among HIV-positive women: Evidence from government facilities in Kenya

PLOS ONE

Dear Dr. Biomndo,

Thank you for submitting your manuscript to PLOS ONE. After careful consideration, we feel that it has merit but does not fully meet PLOS ONE’s publication criteria as it currently stands. Therefore, we invite you to submit a revised version of the manuscript that addresses the points raised during the review process.

Thank you for the time and attention you put into this resubmission. As noted by both reviewers, this submission is greatly improved. There are still a few areas that could be addressed as outlined by Reviewer #1 and Reviewer #2 (who was a new reviewer and noted that a paragraph on the conceptual framing of the study, including why the covariates were selected is needed). Both reviewers have provided very detailed feedback which will continue to improve this manuscript. We look forward to receiving your revision.

We look forward to receiving your revised manuscript.

Kind regards,

Michelle L. Munro-Kramer, PhD, CNM, FNP-BC

Academic Editor

PLOS ONE

Reviewers' comments:

Reviewer's Responses to Questions

**Comments to the Author**

1. If the authors have adequately addressed your comments raised in a previous round of review and you feel that this manuscript is now acceptable for publication, you may indicate that here to bypass the “Comments to the Author” section, enter your conflict of interest statement in the “Confidential to Editor” section, and submit your "Accept" recommendation.

Reviewer #3: (No Response)

Reviewer #4: (No Response)

2. Is the manuscript technically sound, and do the data support the conclusions?

Reviewer #3: Partly

Reviewer #4: Yes

3. Has the statistical analysis been performed appropriately and rigorously? 

Reviewer #3: I Don't Know

Reviewer #4: Yes

4. Have the authors made all data underlying the findings in their manuscript fully available?

Reviewer #3: Yes

Reviewer #4: Yes

5. Is the manuscript presented in an intelligible fashion and written in standard English?

Reviewer #3: Yes

Reviewer #4: Yes

6. Review Comments to the Author

Reviewer #3: This manuscript is an interesting foray into an understudied population. The authors should be commended for their work from the previous draft- it seems much improved. However, there are some methodological questions and framing issues that need attention before the manuscript should be considered for publication.

Line 71: Please expound upon the proposed mechanisms in the literature between IPV and HIV management service uptake

There is a lack of theoretical or conceptual underpinnings here- *why* do the authors proposed IPV being a barrier to ARV uptake? This is not a new concept, so existing literature and concepts just need to be brought into this section and tailored for the study context. The authors briefly mention this in Line 285, but this should be corroboration of an earlier theory-driven hypothesis

Line 122: Did the authors alter the DHS questionnaire to focus on relationship-length violence? Since the DHS measures only past-year and lifetime, alterations must have been made to focus on violence ever being perpetrated by the current partner? If so, this is an interesting innovation and should be discussed further.

Methods Note: The other aspects of the model are not discussed- why did the authors choose the predictor variables they did? What is the theory/conceptual model driving these models?

I am concerned with the “woman is violent” covariate. This implies a bidirectionality of IPV that is not theoretically or conceptually consistent with violence in this context. I recommend removing it or strongly justifying it in the (yet nonexistent) paragraph explaining the other model covariates.

Line 166: Can the authors provide a citation or reference to whether 100 shillings is standard compensation for such a survey? Non-Kenyan audiences may call into the question the coerciveness of this amount without this context

Line 179: “virological” is not a word- should be “viral” suppression

Line 192: Despite the potential bias of small numbers of second-level units, the authors should have performed a sensitivity analysis to determine if results varied widely using a two-level model. The authors should have at least included clinic as a fixed effect at level 1 if not using multilevel modeling to control for the effect of sampling site

Line 210: I assume the authors did not try to fit logistic regression models with such small cell sizes as in Age<20 (n=1). Categories of the predictor variables should be reported in the way they were modeled in the final table

Line 228: Why are variables modeled differently in Table 1 vs Table 2 (e.g. Age modeled as continuous in table 2 but categorical in table 1?). These should be consistent.

Line 214: I recommend using positive verbiage here- i.e. “physical, sexual, and any type of IPV remained significant”

Line 246: citations needed here for the Kenyan context

Line 372: I feel the conclusion needs work. What are the final lessons learned and next steps for this line of inquiry- additional thoughts from the researchers would be welcome here.

Reviewer #4: (No Response)

7. PLOS authors have the option to publish the peer review history of their article (what does this mean?). If published, this will include your full peer review and any attached files.

Reviewer #3: No

Reviewer #4: No

---

## [Author Response · Author response to Decision Letter 1]

8 Feb 2021

Reviewer’s comments

This manuscript is an interesting foray into an understudied population. The authors should be commended for their work from the previous draft- it seems much improved. However, there are some methodological questions and framing issues that need attention before the manuscript should be considered for publication.

1a) Line 71: Please expound upon the proposed mechanisms in the literature between IPV and HIV management service uptake. 

1b) There is a lack of theoretical or conceptual underpinnings here- *why* do the authors proposed IPV being a barrier to ARV uptake? This is not a new concept, so existing literature and concepts just need to be brought into this section and tailored for the study context. The authors briefly mention this in Line 285, but this should be corroboration of an earlier theory-driven hypothesis.

1c) I miss a description of why there might be an association between IPV and adherence in the Introduction.

Thank you for the comments. We decided to combine the three comments because we believe they stem from the missing conceptual framework which we should have included. 

Along the HIV care cascade (i.e. HIV infection, diagnosis, linking to care, retention in care, initiation of ART, and achieving viral suppression), there are several points at which IPV is said to interfere with interventions being successful. Different studies have tried to establish and quantify this interference at different steps of the cascade. For our study, we focus on the last step which is achieving viral suppression through ART adherence. However, we briefly introduce the other cascade points within our introduction section as background information on the relationship between IPV and ART adherence. 

It is true that we failed to include the conceptual framework which could have made this clearer. Therefore in the revised manuscript we included the conceptual framework on ART adherence which guided our research study on Pg. 4 line 70 -73. That is, ‘Alongside treatment regime, provider-patient relationship, clinic setting, and disease characteristics, the conceptual framework on ART adherence identifies sociodemographic drivers as having the potential to influence a patient’s level of medical adherence. Particularly, that there are socio-cultural and interpersonal factors (i.e. partner interference) that physically or psychologically undermine a woman’s ability or motivation to adhere to ART. 

2a) Line 122: Did the authors alter the DHS questionnaire to focus on relationship-length violence? Since the DHS measures only past-year and lifetime, alterations must have been made to focus on violence ever being perpetrated by the current partner? If so, this is an interesting innovation and should be discussed further.

2b) The authors use self-reported lifetime IPV restricted to the current partner. Since they first write that they use the DHS IPV module and then (correctly) state that it measures IPV by current and former partners (page 6-7), the claim that the authors focus on lifetime IPV of the ongoing relationship is confusing. Either the claim is wrong or the authors have modified the DHS IPV module. If they have modified it, this should be clearly stated.

Our study focused on relationship-specific IPV and we did this by recruiting only the women who were currently in an intimate partnership (either married or in a relationship). The DHS questions on IPV are framed as ‘Did your (last) husband/partner ever..’ to capture IPV by current or former partner. To make the questions fit to our focus, we removed the word ‘last’ so that our questions read ‘Did your husband/partner ever..’. We apologise for not clarifying this earlier and we have added a few lines in Pg. 7 from line 129 explaining this.

3) Methods Note: The other aspects of the model are not discussed- why did the authors choose the predictor variables they did? What is the theory/conceptual model driving these models?

Thank you for pointing this out. Within the data collection section, we have rewritten the sentence on the independent variables to state why they were used (Pg. 7 from line 146). We hope that this, together with the now added theoretical framework, will make it easier for the reader to understand why we included these covariates.

4) I am concerned with the “woman is violent” covariate. This implies a bidirectionality of IPV that is not theoretically or conceptually consistent with violence in this context. I recommend removing it or strongly justifying it in the (yet nonexistent) paragraph explaining the other model covariates.

Thank you for the comment. This covariate was added because it is an aspect which the DHS IPV questionnaire measures. Since we believe there was a hypothesis behind adding it in the domestic violence module, were interested in investigating whether it would influence our results. From the results, the variable does not add any strong argument for or against our hypothesis. It also does not stand out as a strong point of discussion. Nevertheless, for the sake of transparency we presented it because it was significant in the bivariate analyses and in some points of the logistic regression. 

 5). Line 166: Can the authors provide a citation or reference to whether 100 shillings is standard compensation for such a survey? Non-Kenyan audiences may call into the question the coerciveness of this amount without this context.

To the best of our knowledge there is no standard compensation for such a survey that we can reference. The research was paid for using limited private funds. After consulting with the AMPATH leadership and the healthcare workers, we agreed that KSh 100 (a little less than 1€ at the time) which was all we could afford, was better than giving nothing. 

With regard to the possible coerciveness of this amount, we describe in the procedure section (Pg. 8 line 166 -174) that consent to participate in the study was sought through explaining the purpose of the study and reassuring the women that participation or non-participation was purely voluntary and would not affect their access to ART. The compensation was offered only after consent was given and the women had completed the questionnaire. Moreover, in terms of purchasing power KSh 100 is unfortunately very low. We would have wished to offer the participants more, had we had proper funding.

6) Line 179: “virological” is not a word- should be “viral” suppression

To the best of our knowledge, virological suppression and viral suppression can be used interchangeably. We also decided to use “virological suppression” as it was already used in other papers published in PlosOne and PlosMedicine, most recently Plymoth M. et al [1] and Hermans, L.E. et al [2].However, we are happy to use the term ‘viral suppression’ instead if it fits better to the discourse of PlosOne. 

7) Line 192: Despite the potential bias of small numbers of second-level units, the authors should have performed a sensitivity analysis to determine if results varied widely using a two-level model. The authors should have at least included clinic as a fixed effect at level 1 if not using multilevel modelling to control for the effect of sampling site

Thank you for highlighting this. To provide the reader with the additional information we have added more details on the analyses we conducted before reaching the decision on our final model on Pg. 10 from line 200. That is, we decided against a multilevel modelling procedure due to the comparatively low intraclass correlation coefficient of ICC=.15 which we derived from an unconditional multilevel logistic regression model. Additionally, with as few as 12 clusters, fixed-effect estimates associated with level-2 predictors could have been severely biased. Adding clinic as an independent variable as part of our sensitivity analysis revealed differences among the clinics but there were no significant changes in the respective model parameter estimates from previous models. 

8) Since data on IPV during the last 12 months are available, which in my view would have been more suitable for the study than lifetime IPV, the authors should at least comment on how the results are altered if they are used.

Noted. Within the data analysis section (Pg. 9 line 184), we have now added a short paragraph explaining that we also ran a second analysis considering only exposure to IPV in the last 12 months. The results of this second analysis have been added in the results section (Pg. 11 line 223-225, and Pg. 12 line 261-267). The findings mirror those of the relationship-specific IPV analysis; i.e. experiencing IPV reduces the odds of achieving optimal ART adherence, while education level and having an HIV-positive partner increased the odds of achieving optimal ART adherence. This further strengthens our rationale that specific IPV incidents may not affect medication adherence, but living in an environment in which IPV occurs does.

9a) Line 210: I assume the authors did not try to fit logistic regression models with such small cell sizes as in Age<20 (n=1). Categories of the predictor variables should be reported in the way they were modelled in the final table.

9b) Line 228: Why are variables modelled differently in Table 1 vs Table 2 (e.g. Age modelled as continuous in table 2 but categorical in table 1?). These should be consistent.

9c) What does marital status, monogamy, polygamy in Table 1 mean? The classification is unclear

Thank you for bringing these points to our attention. We apologise for the oversight and have changed them accordingly (Pg. 11)

10) The authors write as if unadjusted results are reported, see p. 11 lin3-4. However, they are not reported. 

Thank you for pointing this out. We had presented both the unadjusted and adjusted results in our earlier manuscripts, hence the tone in reporting and the references. We have now revised this to fit to what is actually presented in the current manuscript (Pg. 12 from line 251). 

11) Line 214: I recommend using positive verbiage here- i.e. “physical, sexual, and any type of IPV remained significant”

Noted. We changed the text following the reviewer’s suggestion (Pg. 12 line 252-253).

12) Line 246: citations needed here for the Kenyan context

We have added the relevant citations (Pg. 14. Line 289).

13) Typo, p.19, line 6. on should be of.

Noted. We decided to change the statement to ‘A review on programs in sub-Saharan Africa which have implemented IPV screening and counselling services in healthcare settings, reported that the interventions were positively received by both healthcare providers and clients’ (Pg. 19 line 412).

14a) Line 372: I feel the conclusion needs work. What are the final lessons learned and next steps for this line of inquiry- additional thoughts from the researchers would be welcome here. 

14b) “ART programs should consider incorporating basic IPV interventions into regular clinic services and using healthcare providers as information and navigation points for women exposed to IPV” does not follow from the findings. Such a conclusion should be based on evidence of a casual effect of IPV on ART adherence. An association does not have to be due to the impact of IPV on adherence. For example, women who end up with violent men might be different from others in ways not captured by the control variables, making them care less about their health. Moreover, violent men might have characteristics that influence their partners’ adherence. One example is that they are likely to be less faithful and less caring. Thus, a program that reduces IPV might not improve adherence. Although it seems reasonable that IPV actually has a causal effect on adherence, the conclusion drawn from the study should be modified.

Noted. We decided to combine the two points because we believe they are related. 

It is true that our findings show only a negative association and not a causal effect of IPV on ART adherence. From a researcher’s point of view, we consider this a limitation (as stated in the limitation section Pg. 18 line 389). However, from a practitioner’s point of view we still stand by our recommendation to consider introducing brief IPV screening and counselling at clinic level as a preventive measure. As existing literature on ART has established, the factors that influence ART adherence are diverse [3-9]. Therefore, it is true that in the end, any reduction in the effects of IPV may not result in improved ART adherence. Nonetheless, since our findings tell us that exposure to IPV could create a group that is vulnerable to suboptimal ART adherence, we recommend identifying these women who need extra monitoring and support. At the same time, given the high prevalence of IPV and knowing that, in general, IPV has negative effects on women’s health, an intervention would be good for the HIV-positive woman’s overall quality of life. As it stands, the jury is still out on whether IPV screening and counselling at clinical level has direct or indirect benefits on women’s health. We request future researcher to share their findings. 

We have adjusted the future practices and conclusion section to reflect these sentiments (Pg. 19 line 398).

15) A related issue is the consequences of IPV on HIV-infections. The authors describe these in the first two paragraphs of the paper and refer to several papers. However, they miss some important references and the fact that recent evidence indicates that there might not be a causal effect of IPV on HIV-infections among women in Sub-Saharan Africa. Women who live with violent men are primarily infected because violent men are more likely than others to be HIV-infected. See for example, Durevall and Lindskog (2015) and Heise and McGrory (2016).

Thank you for the comment. 

It is true that there are studies that question the bidirectional association between IPV and positive HIV serology. Although from our understanding, the two papers mentioned do not completely rule out the possibility of an association, but rather emphasise the importance of considering the woman’s partner’s sexual behaviour. That is, interventions should also focus on changing men’s sexual behaviour.

Nevertheless, it is true that we did not mention studies that counter the association between IPV and HIV infection. This is because our study focus was not HIV infection, but rather ART adherence. We therefore believe it is more relevant for us to discuss studies which support or reject that IPV has an influence on ART adherence. We understand that this might not have been clear before and we hope that including the theoretical framework helps to clarify it.

16) A drawback of the study is that women who have not entered an ART program or have dropped out are not included in the sample. This should be made clear early on, particularly since it is likely to create a bias that weakens the association. It should also be commented on in Limitations. Now it is mentioned in Conclusion, which is odd.

Noted. In our view, ‘not including women who have not entered the ART program in our sample’ is not a limitation because our aim was to determine IPV’s influence on ART adherence. Our target was therefore HIV-positive women who were already on ART. We agree that IPV could be a factor that prevents women from initiating ART (HIV care cascade step 3) and we have referenced, within the introduction section, some studies that highlight this (Pg. 4 from line 64). 

With regard to not including women who have dropped out of the ART program, we have moved the mentioned drawback to the limitation section (Pg. 18 line 386). We have also retained it in the conclusion section because we believe it offers a point for future research.

---

## [Decision Letter · Decision Letter 2]

3 Mar 2021

PONE-D-20-21450R2

Intimate partner violence is a barrier to antiretroviral therapy adherence among HIV-positive women: Evidence from government facilities in Kenya

PLOS ONE

Dear Dr. Biomndo,

Thank you for submitting your manuscript to PLOS ONE. After careful consideration, we feel that it has merit but does not fully meet PLOS ONE’s publication criteria as it currently stands. Therefore, we invite you to submit a revised version of the manuscript that addresses the points raised during the review process.

Thank you for your edits to this manuscript. As noted by both reviewers, the reviews addressed the majority of the reviewers' comments. There are still a few minor outstanding comments that should be addressed before publication. We look forward to receiving the revision.

We look forward to receiving your revised manuscript.

Kind regards,

Michelle L. Munro-Kramer, PhD, CNM, FNP-BC

Academic Editor

PLOS ONE

Journal Requirements:

Reviewers' comments:

Reviewer's Responses to Questions

**Comments to the Author**

1. If the authors have adequately addressed your comments raised in a previous round of review and you feel that this manuscript is now acceptable for publication, you may indicate that here to bypass the “Comments to the Author” section, enter your conflict of interest statement in the “Confidential to Editor” section, and submit your "Accept" recommendation.

Reviewer #3: (No Response)

Reviewer #4: All comments have been addressed

2. Is the manuscript technically sound, and do the data support the conclusions?

Reviewer #3: Yes

Reviewer #4: Yes

3. Has the statistical analysis been performed appropriately and rigorously? 

Reviewer #3: Yes

Reviewer #4: Yes

4. Have the authors made all data underlying the findings in their manuscript fully available?

Reviewer #3: Yes

Reviewer #4: Yes

5. Is the manuscript presented in an intelligible fashion and written in standard English?

Reviewer #3: Yes

Reviewer #4: Yes

6. Review Comments to the Author

Reviewer #3: The authors have responded well to many of the previous reviewers’ comments. I believe that the manuscript can be forwarded to the editor for final approval, provided some small changes are made. Most significant among these are the expansion of the conceptual model in the Introduction and the explanation of a few key findings in the Discussion.

Introduction:

• The conceptual model is still lacking in my opinion. There are three distinct ways in which IPV may lead to negative uptake of health services, including ART. I refer the authors to the seminal WHO conceptual model contained in the 2013 Global and regional estimates of violence against women: prevalence and health effects of intimate partner violence and non-partner sexual violence report.

Methods:

• Although I personally disagree with the decision not to use multilevel modeling, the authors justify this decision well and there is no reason to suspect this decision meaningfully changed the results or their interpretation.

• I recommend moving the justification for using the stepwise regression approach to the Methods section from the Limitations section

Discussion:

• What are the authors’ hypotheses regarding the mechanisms or reasons behind why women who fought back and experience controlling behavior have reduced ART adherence? This should be discussed.

Reviewer #4: The revised version of the paper is fine and I have only some very minor comments.

On line 106-108 the authors write that “We set a minimum duration of six months since beginning ART to recruit women who were accustomed to the ART routine, in an effort to reduce factors affecting adherence that may be related to the initiation stages of ART.” I would like to have some more arguments. Why would using a much smaller duration affect the results? There would be more noise, but you would have more observations and I assume that the correlation with IPV would not be affected. Did you test using a shorter duration?

I do not understand this part. “We also did not include clinics as a fixed effect because this would lead to empty cells due to limited sample size. However, we included clinic as an independent variable as part of our sensitivity analysis” Fixed effects can be described as a dummy for each unit (clinic), which is what is used in the sensitivity analysis.

The authors have carried out estimations that are not reported, but they never state that they are not reported, such as the analysis of women experiencing IPV in the last 12 months. I think these results, and the sensitivity analysis, should be included in the on-line appendix, which now only contains one table.

The authors write that “However, this finding should be considered cautiously, since the tertiary education subgroup had only 49 respondents, which led to small cell numbers for some predictor combinations.” But 49 observations sound sufficient. And why would the cell for the tertiary education subgroup have fewer than 49 observations in some four reported regressions?

In table 2, the p-value for Partner’s HIV status in model 3 and 4 are very different even though the estimates are very similar: 1.60 0.97 – 2.62 0.64; 1.60 0.98 – 2.63 .062. Typo?

7. PLOS authors have the option to publish the peer review history of their article (what does this mean?). If published, this will include your full peer review and any attached files.

Reviewer #3: No

Reviewer #4: **Yes: **Dick Durevall

---

## [Author Response · Author response to Decision Letter 2]

19 Mar 2021

Reviewer’s comments

Reviewer #3

The authors have responded well to many of the previous reviewers’ comments. I believe that the manuscript can be forwarded to the editor for final approval, provided some small changes are made. Most significant among these are the expansion of the conceptual model in the Introduction and the explanation of a few key findings in the Discussion.

Introduction:

• The conceptual model is still lacking in my opinion. There are three distinct ways in which IPV may lead to negative uptake of health services, including ART. I refer the authors to the seminal WHO conceptual model contained in the 2013 Global and regional estimates of violence against women: prevalence and health effects of intimate partner violence and non-partner sexual violence report.

Thank you for highlighting this. We have decided to add a paragraph on Pg.5 from line 88. In this paragraph we describe the different pathways in which IPV directly and indirectly affects women’s health and how this could link to ART adherence. We have taken into account the abovementioned conceptual model. 

Methods:

• Although I personally disagree with the decision not to use multilevel modeling, the authors justify this decision well and there is no reason to suspect this decision meaningfully changed the results or their interpretation.

Noted

• I recommend moving the justification for using the stepwise regression approach to the Methods section from the Limitations section.

Done. Please see Pg. 10 line 206

Discussion:

• What are the authors’ hypotheses regarding the mechanisms or reasons behind why women who fought back and experience controlling behavior have reduced ART adherence? This should be discussed.

Noted. Our results show that adjusting for controlling behaviour, women who fought back during an IPV encounter were more likely to report reduced odds of ART adherence than those who are never violent. Violence by women who are exposed to IPV is not uncommon as studies show it is one of the means through which some women defend themselves and their children [1-3]. Since we know that women who are exposed to IPV adopt different physical and nonphysical ways of protecting themselves, we cannot confidently draw any conclusions about the mentioned sub-group of women without having further assessments. 

Our results suggest that the act of fighting back during an IPV encounter has no effect on ART adherence. This is evident by the fact that there is no effect when adjusted for physical and sexual IPV. However, a negative effect is seen when adjusted for controlling behaviour, which speaks more to the environment. We interpret this as being reflective of the underlying and underestimated detrimental effect controlling behaviour has, and encourage further research. One way would be through assessing the home environment in which the controlling behaviour occurs. Moreover, we believe that the negative effect on optimal adherence which is seen when adjusted for emotional IPV and controlling behaviour, furthers our rationale that it is not the IPV act itself, but rather living within an environment where IPV occurs, that affects a woman’s ability or motivation to adhere to ART.

We nevertheless realise our error in not making this clear in the beginning and have added a few lines explaining this in Pg 18 from line 343.

Reviewer #4

The revised version of the paper is fine and I have only some very minor comments.

a) On line 106-108 the authors write that “We set a minimum duration of six months since beginning ART to recruit women who were accustomed to the ART routine, in an effort to reduce factors affecting adherence that may be related to the initiation stages of ART.” I would like to have some more arguments. Why would using a much smaller duration affect the results? There would be more noise, but you would have more observations and I assume that the correlation with IPV would not be affected. Did you test using a shorter duration?

Thank you for this comment. We would like to address this in two ways. Firstly, we did not test using a shorter duration because it would have meant that we needed to also measure factors that are negatively associated with ART initiation. This includes but is not limited to: the woman’s knowledge and understanding of HIV/ART, the woman’s denial, fears or expectations of HIV/ART, challenges to the practical demands of treatment (e.g. scheduling, reluctance to start lifelong treatment), health worker-patient relations, adverse drug reaction, and regimen adjustments. Additionally, women who attend the ART program are referred through Home Based Counselling and Testing (HBCT), Voluntary Counselling and Testing (VCT) or Provider Initiated Testing and Counselling (PITC). This already puts them at different levels in terms of acceptance of HIV diagnosis, CD4 count levels, and their motivation to take up or adhere to treatment. All these together with the IPV related variables would inflate the number of independent variables we assess in our sample of 408 women.

Secondly, from a practitioner’s perspective, because of the aforementioned factors, we expect some instability in ART adherence in the first few months. According to the national guidelines for antiretroviral therapy in Kenya [4], it is after six months following initiation of ART that a patient is expected to have sufficient knowledge of HIV/ART and the importance of adherence, and is therefore expected to be compliant. Before six months, the guidelines recommend frequent appointments (two weeks, monthly) to assess proper administration and storage of the medication, to assess patient’s clinical progress, and to counsel on adherence. It is also during this duration that the patient is closely monitored for adverse drug reactions, drug resistance, and adjustments to the treatment regimen is made. At six months, a CD4 count is recommended as a follow up to the baseline CD4 count which was conducted when the HIV status was confirmed. We therefore expected that the women who participated in our survey had sufficient knowledge of HIV, medication compliance, and the importance and benefits of ART adherence. 

The number of observations were not affected since we had no trouble achieving the target sample size using the inclusion criteria. Nonetheless, your comment has made us realise that this is information that may not be clear to the readers and so we have added a brief explanation on Pg. 6 from line 113. 

b) I do not understand this part. “We also did not include clinics as a fixed effect because this would lead to empty cells due to limited sample size. However, we included clinic as an independent variable as part of our sensitivity analysis” Fixed effects can be described as a dummy for each unit (clinic), which is what is used in the sensitivity analysis.

Noted. We did not include clinics as a fixed effect in our main analysis because adding twelve additional dummy coded variables to the analysis would have led to predictor combinations with very few to zero observations. In this case, the specified model would have smoothed over the points where only limited data is available, which in turn would have led to a decrease in confidence in the respective models’ estimators. We have added this explanation on Pg 10 line 216.

c)The authors have carried out estimations that are not reported, but they never state that they are not reported, such as the analysis of women experiencing IPV in the last 12 months. I think these results, and the sensitivity analysis, should be included in the on-line appendix, which now only contains one table.

Noted. We have included the tables showing the results of the multiple regression models using only women who experienced IPV in the last 12 months, and the one including clinics is the Supplementary Information.

d) The authors write that “However, this finding should be considered cautiously, since the tertiary education subgroup had only 49 respondents, which led to small cell numbers for some predictor combinations.” But 49 observations sound sufficient. And why would the cell for the tertiary education subgroup have fewer than 49 observations in some four reported regressions?

Thank you for the comment. Out of all the women we randomly surveyed only 49 had tertiary education. When we combined education level with the other predictor variables there were instances where the cells were empty or had very few observations. For example, there were no women with tertiary education who were in a relationship and had experienced physical IPV. Only two women with tertiary education were in a polygamous marriage. 

Nonetheless, we decided to rephrase the previous statement to read ‘..However, it should be noted that the tertiary education subgroup had only 49 respondents, which led to small cell or no numbers for some predictor combinations’ (Pg.18. line 339)

e) In table 2, the p-value for Partner’s HIV status in model 3 and 4 are very different even though the estimates are very similar: 1.60 0.97 – 2.62 0.64; 1.60 0.98 – 2.63 .062. Typo?

Thank you for bringing this to our attention. It was indeed a typo and we have corrected it.

References

1. Swan SC, Snow DL. The development of a theory of women's use of violence in intimate relationships. Violence against women. 2006 Nov;12(11):1026-45. PubMed PMID: WOS:000241242700005. English.

2. Downs WR, Rindels B, Atkinson C. Women's use of physical and nonphysical self-defense strategies during incidents of partner violence. Violence against women. 2007 Jan;13(1):28-45. PubMed PMID: WOS:000242973100003. English.

3. Denson TF, O'Dean SM, Blake KR, Beames JR. Aggression in Women: Behavior, Brain and Hormones. Front Behav Neurosci. 2018 May 2;12. PubMed PMID: WOS:000431189600001. English.

4.-NASCOP. Guidelines for Antiretroviral Therapy in Kenya. Nairobi Kenya: National AIDS and STI Control Program, 2011.

---

## [Editor Report · Decision Letter 3]

26 Mar 2021

Intimate partner violence is a barrier to antiretroviral therapy adherence among HIV-positive women: Evidence from government facilities in Kenya

PONE-D-20-21450R3

Dear Dr. Biomndo,

We’re pleased to inform you that your manuscript has been judged scientifically suitable for publication and will be formally accepted for publication once it meets all outstanding technical requirements.

Kind regards,

Michelle L. Munro-Kramer, PhD, CNM, FNP-BC

Academic Editor

PLOS ONE

Additional Editor Comments (optional):

Thank you for your continued work attending to the reviewers' comments. The clarity of the manuscript has been greatly improved over the course of revisions. I am happy to accept this manuscript for submission.
---

## [Editor Report · Acceptance letter]

30 Mar 2021

PONE-D-20-21450R3 

Intimate partner violence is a barrier to antiretroviral therapy adherence among HIV-positive women: Evidence from government facilities in Kenya 

Dear Dr. Biomndo:

I'm pleased to inform you that your manuscript has been deemed suitable for publication in PLOS ONE. Congratulations! Your manuscript is now with our production department. 

Kind regards, 

on behalf of

Dr. Michelle L. Munro-Kramer 

Academic Editor

PLOS ONE